



**Ozone Production and Its Sensitivity to NOx and VOCs: Results from the DISCOVER-AQ**
**Field Experiment, Houston 2013**
Gina M. Mazzuca[1], Xinrong Ren[1,2,*], Christopher P. Loughner[2,3,4], Mark Estes[5], James H.
Crawford[6], Kenneth E. Pickering[1,4], Andrew J. Weinheimer[7], and Russell R. Dickerson[1]
[1]Department of Atmospheric and Oceanic Science, University of Maryland, College Park, MD
20742, USA
[2]Air Resources Laboratory, National Oceanic and Atmospheric Administration, College Park,
MD 20740, USA
[3]Earth System Science Interdisciplinary Center, University of Maryland, College Park, MD
20740, USA
[4]NASA Goddard Space Flight Center, Greenbelt, MD 20771, USA
[5]Texas Commission on Environmental Quality, Austin, TX 78711, USA
[6]NASA Langley Research Center, Hampton, VA 23681, USA
[7] National Center for Atmospheric Research, Boulder, CO 80307, USA
*Correspondence to: X. Ren (ren@umd.edu)
**Abstract** An observation-constrained box model based on the Carbon Bond mechanism, Version
5 (CB05), was used to study photochemical processes along the NASA P-3B flight track and
spirals over eight surface sites during the September 2013 Houston, Texas deployment of the
NASA DISCOVER-AQ campaign. Data from this campaign provided an opportunity to examine
and improve our understanding of atmospheric photochemical oxidation processes related to the
formation of secondary air pollutants such as ozone ($O_3$). $O_3$ production and its sensitivity to
$NO_x$ and VOCs were calculated at different locations and times of day. Ozone production
efficiency (OPE), defined as the ratio of the ozone production rate to the $NO_x$ oxidation rate, was
calculated using the observations and the simulation results of the box and Community
Multiscale Air Quality (CMAQ) models. Correlations of these results with other parameters,
such as radical sources and $NO_x$ mixing ratio, were also evaluated. It was generally found that $O_3$
production tends to be more VOC sensitive in the morning along with high ozone production
rates, suggesting that control of VOCs may be an effective way to control $O_3$ in Houston. In the





afternoon, $O_3$ production was found to be mainly $NO_x$ sensitive with some exceptions. $O_3$
production at near major emissions sources such as Deer Park was mostly VOC sensitive for the
entire day, other urban areas near Moody Tower and Channelview were VOC sensitive or in the
transition regime, and areas farther from downtown Houston such as Smith Point and Conroe
were mostly $NO_x$ sensitive for the entire day. It was also found that the control of $NO_x$ emissions
has reduced $O_3$ concentrations over Houston, but led to larger OPE values. The results from this
work strengthen our understanding of $O_3$ production; they indicate that controlling $NO_x$
emissions will provide air quality benefits over the greater Houston metropolitan area in the long
run, but in selected areas controlling VOC emissions will also be beneficial.

**Keywords** Ozone production; Houston; DISCOVER-AQ

**1. Introduction**

Understanding the non-linear relationship between ozone production and its precursors is

critical for the development of an effective ozone ($O_3$) control strategy. Despite great efforts
undertaken in the past decades to address the problem of high ozone concentrations, our
understanding of the key precursors that control tropospheric ozone production remains
incomplete and uncertain [Molina and Molina, 2004; Xue et al., 2013]. Atmospheric ozone
levels are determined by emissions of ozone precursors, atmospheric photochemistry, and
transport  [Jacob, 1999; Xue et al., 2013]. A major challenge in regulating ozone pollution lies in
comprehending its complex and non-linear chemistry with respect to ozone precursors, i.e.,
nitrogen oxides ($NO_x$) and volatile organic compounds (VOCs) that varies with time and location
(Figure 1). Understanding of the non-linear relationship between ozone production and its
precursors is critical for the development of an effective ozone control strategy.

Sensitivity of ozone production to $NO_x$ and VOCs represents a major uncertainty for

oxidant photochemistry in urban areas [Sillman et al., 1995; 2003]. In urban environments,
ozone is formed through photochemical processes when its precursors $NO_x$ and VOCs are
emitted into the atmosphere from many sources. Depending on physical and chemical conditions,
the production of ozone can be either $NO_x$-sensitive or VOC-sensitive due to the complexity of
these photochemical processes. Therefore, effective ozone control strategies rely heavily on the
accurate understanding of how ozone responds to reduction of $NO_x$ and VOC emissions, usually



simulated by photochemical air quality models [e.g., Sillman et al., 2003; Lei et al., 2004; Mallet
and Sportisse, 2005; Li et al, 2007; Chen et al., 2010; Tang et al., 2010; Xue et al., 2013;
Goldberg et al., 2016]. However, those model-based studies have inputs or parameters subject to
large uncertainties that can affect not only the simulated levels of ozone but also the ozone
dependence on its precursors.
There are a limited number of observation-based studies on ozone production and its
sensitivity to $NO_x$ and VOCs. Using in-situ aircraft observations, Kleinman et al. [2005a] studied
five U.S. cities and found that ozone production rates vary from nearly zero to 155 ppb hr$^{-1}$ with
differences depending on precursor concentrations $NO_x$, and VOCs.  They also found that in
Houston, $NO_x$ and light olefins are co-emitted from petrochemical facilities leading to the
highest ozone production of the five cities [Kleinman et al., 2005a]. Using the data collected at a
single surface location during the Study of Houston Atmospheric Radical Precursors (SHARP) in
spring 2009, the temporal variation of $O_3$ production was observed: VOC-sensitive in the early
morning and $NO_x$-sensitive for most of the afternoon [Ren et al., 2013]. This is similar to the
behavior observed in two previous summertime studies in Houston: the Texas Air Quality Study
in 2000 (TexAQS 2000) and the TexAQS II Radical and Aerosol Measurement Project in 2006
(TRAMP 2006) [Mao et al., 2010; Chen et al., 2010]. In a more recent study using measurements
in four cities in China, ozone production was found to be in a VOC-sensitive regime in both
Shanghai and Guangzhou, but in a mixed regime in Lanzhou [Xue et al., 2013].  More
investigations of spatial and temporal variations of ozone production and its sensitivity to $NO_x$
and VOCs are thus needed to provide a scientific basis to develop a non-uniform emission
reduction strategy for $O_3$ pollution control in urban areas like Houston.
During the Deriving Information on Surface Conditions from COlumn and VERtically
Resolved Observations Relevant to Air Quality (DISCOVER-AQ) campaign in Houston in
September 2013, a comprehensive suite of measurements were collected from various platforms
including the National Aeronautics and Space Administration (NASA) P-3B and B-200 aircraft,
ground surface sites, and mobile laboratories [DISCOVER-AQ whitepaper].  In-situ
measurements on the NASA P-3B directly related to satellite observations of air quality include
ozone ($O_3$), nitrogen dioxide ($NO_2$), formaldehyde (HCHO), and aerosol optical and
microphysical properties. Additional critical variables needed for retrievals and data
interpretation were also measured including atmospheric state (temperature, pressure, wind speed



and wind direction), water vapor ($H_2O$), carbon monoxide (CO), methane ($CH_4$), carbon dioxide
($CO_2$), nitric oxide (NO), the other components of reactive nitrogen, and aerosol inorganic and
organic composition.

Eight surface monitoring stations were selected where the P-3B conducted vertical spirals

(Figure 2). These monitoring stations provided in situ observations of trace gases ($O_3$, CO, NO,
$NO_y$, $SO_2$), and at a subset of these stations aerosol lidar observations, $NO_2$ columns, and balloon
soundings of $O_3$, $NO_2$, $NO_x$ and water vapor were conducted. The eight surface sites (Smith
Point, Galveston, Manvel Croix, Deer Park, Channelview, Conroe, West Houston, and Moody
Tower) were chosen for the deployment with regard to the presence or absence of
complementary chemical and meteorological measurements; the strength and likely impact of
nearby point and mobile emission sources; the topography, height, and extent of nearby
structures and vegetation; and any characteristic which might render the site physically or
chemically unrepresentative of the surrounding area [DISCOVER-AQ whitepaper].

## 108   2. Methods

### 109   2.1 Ozone production Scenarios and Sensitivity

During the day, the photochemical $O_3$ production rate is essentially the production rate of

$NO_2$ molecules from $HO_2$ + NO and $RO_2$ + NO reactions [Finlayson-Pitts and Pitts, 2000]. The
net instantaneous photochemical $O_3$ production rate, P($O_3$), can be written approximately as the
following equation:

$$P(O_3) = k_{HO_2+NO}[HO_2][NO] + \sum k_{RO_{2i}+NO}[RO_{2i}][NO] - k_{OH+NO_2+M}[OH][NO_2][M] - P(RONO_2)$$
$$-k_{HO_2+O_3}[HO_2][O_3] - k_{OH+O_3}[OH][O_3] - k_{O(^1D)+H_2O}[O(^1D)][H_2O] - L(O_3 + alkenes) \quad (1)$$

where, *k term*s are the reaction rate coefficients; $RO_{2i}$ is the individual organic peroxy radicals.
The negative terms in Eq. (1) correspond to the reaction of OH and $NO_2$ to form nitric acid, the
formation of organic nitrates, P($RONO_2$), the reactions of OH and $HO_2$ with $O_3$, the photolysis of
$O_3$ followed by the reaction of $O(^1D)$ with $H_2O$, and $O_3$ reactions with alkenes. Ozone is
additionally destroyed by dry deposition.

The dependence of $O_3$ production on $NO_x$ and VOCs can be categorized into two typical

scenarios: $NO_x$ sensitive and VOC sensitive. The method proposed by Kleinman [2005b] was
used to evaluate the $O_3$ production sensitivity using the ratio of $L_N/Q$, where $L_N$ is the radical
loss via the reactions with $NO_x$ and Q is the total primary radical production. Because the radical





production rate is approximately equal to the radical loss rate, this $L_N/Q$ ratio represents the
fraction of radical loss due to $NO_x$. It was found that when $L_N/Q$ is significantly less than 0.5, the
atmosphere is in a $NO_x$-sensitive regime, and when $L_N/Q$ is significantly greater than 0.5, the
atmosphere is in a more VOC-sensitive regime [Kleinman et al., 2001; Kleinman, 2005b]. Note
that the contribution of organic nitrates impacts the cut-off value for $L_N/Q$ to determine the ozone
production sensitivity to $NO_x$ or VOCs and this value may vary slightly around 0.5 in different
environments [Kleinman, 2005b].

**2.2 Box Model Simulations**

An observation-constrained box model with the Carbon Bond Mechanism Version 2005

(CB05) was used to simulate the oxidation processes in Houston during DISCOVER-AQ.
Measurements made on the P-3B were used as input to constrain the box model. From the box
model results, the ozone production rate and its sensitivity to $NO_x$ and VOCs were calculated
allowing us to calculate ozone production efficiency at different locations and at different times
of day.

CB05 is a well-known chemical mechanism that has been actively in use in research and

regulatory applications [Yarwood et al., 2005]. CB05 is an updated version of CB4. In contrast
to the previous version, (1) inorganic reactions are extended to simulate remote to polluted urban
conditions; and (2) two extensions are available to be added to the core mechanism for modeling
explicit species and reactive chlorine chemistry. Organic species are lumped according to the
carbon bond approach, that is, bond type, e.g., carbon single bond and double bond. Reactions
are aggregated based on the similarity of carbon bond structure so that fewer surrogate species
are needed in the model. Some organics (e.g., organic nitrates and aromatics) are lumped. The
original mechanism was used while kinetic data were updated based on the most recent chemical
kinetic data evaluations [e.g., Atkinson et al., 2004; 2006; 2007; 2008; Sander et al, 2011]. The
lifetime of alkyl nitrates is too long in CB05 and has been corrected in CB6r2 [Canty et al.,
2015], but this should have minimal impact on our findings because the model is constrained to
observations as indicated below.

The box model was run using measurements, including long-lived inorganic and organic

compounds and meteorological parameters (temperature, pressure, humidity, and photolysis
frequencies), from the NASA P-3B. One-minute archived data were used as model input



(available at http://www-air.larc.nasa.gov/missions/discover-aq/discover-aq.html). The model
ran for 24 hours for each data point to allow most calculated reactive intermediates to reach
steady state, but short enough to prevent the buildup of secondary products. A deposition lifetime
of two days was assumed for all calculated species to avoid unexpected accumulation of these
species in the model. At the end of 24 hours, the model generated time series of OH, $HO_2$, $RO_2$,
and other reactive intermediates. The box model covered the entire P-3B flight track during
DISCOVER-AQ, including the eight science sites where the P-3B conducted spirals. Note that
unlike a three-dimensional chemical transport model, the zero-dimensional box model
simulations did not include advection and emissions, although advection and emissions are
certainly important factors for the air pollution formation. Because all of the long-lived radical
and $O_3$ precursors were measured and used to constrain the box model calculations, the
advection and emissions can be neglected for the calculated radicals and their production and
loss rates. The box model analysis is necessary for ozone production and its sensitivity to $NO_x$
and VOCs because the box model was constrained to measured species (e.g., NO, $NO_2$, CO,
HCHO, etc.) and meteorological parameters (e.g., photolysis frequencies) that are essential to
calculate ozone production rates. Even though there is good agreement in general between the
box model and the 3D model, there are still some differences between the measurements and the
output from the 3D model, e.g., NOx, CO, HCHO and photolysis frequencies.

**174 2.3 WRF-CMAQ Model Simulations**

The WRF model was run from 18 August 2013 to 1 October 2013 with nested domains
with horizontal resolutions of 36, 12, 4, and 1 km and 45 vertical levels. This work utilized
results from the 4 km domain. The modeling domains are shown in Figures 3 and 4. WRF was
run straight through (i.e., was not re-initialized at all) using an iterative technique developed at
the EPA and described in Appel et al. (2014). Observational and analysis nudging were
performed on all domains. Model output was saved hourly for the 36 and 12 km domains, every
20 minutes for the 4 km domain, and every 5 minutes for the 1 km domain. WRF and CMAQ
configuration options and inputs are shown in Table 1.
WRF model results were used to drive the CMAQ model offline. The CMAQ model was
run with the process analysis tool to output ozone production rate ($P(O_3)$), ozone loss rate
($L(O_3)$), and net ozone production rate (net $P(O_3)$) as well as ozone production efficiency (OPE).




## 3. RESULTS

### 3.1 Photochemical $O_3$ Production Rate, Sensitivity, and Diurnal Variations

Figure 5 shows the net ozone production rate, net $P(O_3)$, calculated using the box model results along the P-3B flight track for all flight days during the Houston deployment. There are several $P(O_3)$ hotspots over the Houston Ship Channel located to the east/southeast of downtown Houston as well as downwind, over Galveston Bay. This is expected because of large emissions of $NO_x$ and VOCs from the Houston Ship Channel, where the highest $P(O_3)$ was observed – up to ~140 ppbv hr$^{-1}$. $P(O_3)$ values up to ~80-90 ppbv hr$^{-1}$ were observed over Galveston Bay, mainly on September 25, 2013, consistent with high ozone levels observed cross the Houston area on that day.

Figure 6 shows the indicator $L_N/Q$ of ozone production sensitivity along the P-3B flight track for all flight days during the Houston deployment. $P(O_3)$ was mainly VOC-sensitive over the Houston Ship Channel and its surrounding urban areas due to large $NO_x$ emissions. Over areas away from the center of the city with relatively low $NO_x$ emissions, $P(O_3)$ was usually NOx-sensitive. Vertical profiles of $P(O_3)$, $L(O_3)$, and net ozone production calculated using the box model results (Figure 7) show that:

(1) $RO_2 + NO$ makes about the same amount of $O_3$ as $HO_2 + NO$ in the model;

(2) $O_3$ photolysis followed by $O(^1D)+H_2O$ is a dominant process for the photochemical ozone loss;

(3) the maximum net $P(O_3)$ appeared near the surface below 1 km.

In the diurnal variations of $P(O_3)$, a broad peak in the morning with significant $P(O_3)$ in the afternoon was obtained on ten flight days during DISCOVER-AQ in Houston (Figure 8). High $P(O_3)$ mainly occurred with $L_N/Q > 0.5$ (i.e., in the VOC sensitive regime). The diurnal variation of $L_N/Q$ indicates that $P(O_3)$ was mainly VOC sensitive in the early morning and then transitioned towards the $NO_x$ sensitive regime later in the day (Figure 9). High $P(O_3)$ in the morning was mainly associated with VOC sensitivity due to high $NO_x$ levels in the morning (points in the red circle in Figure 9). Although $P(O_3)$ was mainly $NO_x$ sensitive in the afternoon between 12:00 and 17:00 Central Standard Time, CST (UTC-5), there were also periods and locations when $P(O_3)$ was VOC sensitive, e.g., the points above the red line in Figure 8.



Diurnal variations of ozone production rate at eight individual locations where the P-3B
conducted vertical spirals show that the ozone production is greater than 10 ppb hr$^{-1}$ on average
at locations with high NO$_x$ and VOC emissions such as Deer Park, Moody Tower and
Channelview, while at locations away from the urban center with lower emissions such as
Galveston, Smith Point, and Conroe, the ozone production usually averaged less than 10 ppb hr$^{-1}$
(Figure 10). The dependence of P(O$_3$) on the NO mixing ratio ([NO]) shows that when [NO] is
less than ~1 ppbv, ozone production increases as the [NO] increases, i.e., P(O$_3$) is in NOx
sensitive regime. When the NO mixing ratio is greater than ~1 ppbv, ozone production levels off,
i.e., P(O$_3$) is in a NOx saturated regime (Figure 11). It was also found that at a given NO mixing
ratio, a higher production rate of HO$_x$ results in a higher ozone production rate. Diurnal
variations of the indicator of ozone production sensitivity to NO$_x$ and VOCs, L$_N$/Q, at eight
individual locations where the P-3B conducted vertical spirals show that (1) at Deer Park, P(O$_3$)
was mostly VOC sensitive for the entire day; (2) at Moody Tower and Channelview, P(O$_3$) was
VOC sensitive or in the transition regime; and (3) at Smith Point and Conroe, P(O$_3$) is mostly
NOx sensitive for the entire day; and Galveston, West Houston, and Manvel Croix were VOC
sensitive only in the early morning (Figure 12).

**233    3.2 Ozone Production Efficiency**

Ozone production efficiency (OPE) is defined as the number of molecules of oxidant Ox
(= O$_3$ + NO$_2$) produced photochemically when a molecule of NO$_x$ (= NO + NO$_2$) is oxidized. It
conveys information about the conditions under which O$_3$ is formed and is an important
parameter to consider when evaluating impacts from NO$_x$ emission sources [Kleinman et al.,
2002]. The OPE can be deduced from atmospheric observations as the slope of a graph of O$_x$
concentration versus the concentration of NO$_x$ oxidation products. The latter quantity is denoted
as NO$_z$ and is commonly measured as the difference between NO$_y$ (sum of all odd-nitrogen
compounds) and NO$_x$, i.e. NO$_z$ = NO$_y$ - NO$_x$.
Figure 13 shows the photochemical oxidant O$_x$ as a function of NO$_z$ during DISCOVER-
AQ in Houston in 2013. The two data sets plotted here were collected on September 25 and 26,
when high ambient ozone concentrations were observed, and for the data collected during all
other flights. Note that the slopes obtained from these two data sets are essentially the same and
an average OPE of ~8 is derived from the observations, meaning that 8 molecules of ozone were





produced when one molecule of $NO_x$ was consumed. Even though higher ozone concentrations
were observed on September 25 and 26, the OPE on these two days are not different from those
in other flights, indicating the ozone event on these two days was not caused by a higher OPE,
but mainly, by higher concentrations of ozone precursors (and thus higher ozone production rates)
and background ozone as indicated by the intercepts in the regression of the two data sets in
Figure 13**.** The high ozone observed on those days could also be due to slower ventilation and
different meteorological conditions such as a lower boundary layer height, northerly transport,
stagnant conditions from the high-pressure system, and the bay and gulf breezes.

The OPE value during DISCOVER-AQ in Houston in 2013 is greater than the average

OPE value (5.9±1.2) obtained during the Texas Air Quality Study in 2006 (TexAQS2006)
[Neuman et al., 2009]. One possible reason for this increased OPE is the continuous reduction in
$NO_x$ emissions in Houston between 2006 and 2013 pushed $NO_x$ levels closer to 1 ppbv in 2013,
thus OPE increased since OPE increases as $NO_x$ decreases when the $NO_x$ level is greater than ~1
ppbv (Figure 14). This OPE value is about a factor of 1.5 to 2 higher than the OPEs obtained in
the DISCOVER-AQ 2011 study in Maryland ranging from 4 to 5.5 (Ren, X., unpublished data),
due to higher photochemical reactivity in Houston.

When calculating ozone production efficiency, it is important to know whether there is

substantial loss of nitric acid ($HNO_3$), because it can affect the OPE by reducing the $NO_z$
[Trainer et al., 1993; 2000; Neuman et al., 2009] and thus bias the OPE high. The derived OPE
in Figure 13 is only valid when there is minimum loss of $NO_z$ (especially $HNO_3$) from the source
region to the point of observations. Neuman et al. [2009] found that $\Delta CO/\Delta NO_y$, i.e., the slope in
a CO versus $NO_y$ plot, is an indicator for distinguishing plumes with efficient $O_3$ formation from
plumes with similarly high $O_3$ to $NO_x$ oxidation products correlation slopes caused by variable
mixing of aged polluted air depleted in $HNO_3$. A typical $\Delta CO/\Delta NO_y$ ranges from ~40 in
background air to ~4-7 in fresh emission plumes in Houston [Neuman et al., 2009]. The
$\Delta CO/\Delta NO_y$ was examined at different times of the day on September 25 and 26. The results
indicate that the $\Delta CO/\Delta NO_y$ was about 6.2 (Figure 15a) throughout the day with variation
between 6.0 and 7.0 (Figure 15). This demonstrates that the observed $O_3$ formation was from
fresh plumes and was not caused by variable mixing of aged polluted air depleted in $HNO_3$.

Using both the box model and CMAQ model results, OPE can also be calculated

according to its definition, i.e., the net ozone formation rate divided by of the formation rate of



$NO_z$. Net $P(O_3)$ was calculated using Eq. (1), while the $NO_z$ formation rate is the sum of $HNO_3$
and organic nitrate formation rates. The agreement between the box model-derived and the
CMAQ-derived OPEs is very good, with the mean OPEs of 14.8±7.4 in the box model and
16.6±8.1 in the CMAQ model. The dependence of OPE on $NO_x$ is also similar for both the box
and CMAQ models (Figure 14). On average, the maximum of OPE appears at a $NO_x$ level
around 1 ppbv. With the $NO_x$ level below 1 ppbv, OPE increases as the $NO_x$ level increases,
while with the $NO_x$ level above 1 ppbv, OPE decreases as the $NO_x$ level increases (Figure 14).

The OPE values calculated using the CMAQ and box model are greater than the values

derived from the observations using the slope in the scatter plot of Ox versus $NO_z$ in Figure 13.
This is expected because in the calculation of OPE using the box and CMAQ model results, a
few ozone loss processes such as ozone dry deposition and horizontal/vertical dispersion were
not considered. This could result in higher calculated ozone production rates using the model
results.

Spatial variations of OPE demonstrate that except for a few hotspots over Downtown

Houston and the Houston Ship Channel, most large OPEs appear away from the urban center,
e.g., the northwest and southeast of the area, while in areas with high $NO_x$ emissions close to the
urban center lower OPEs were generally observed (Figure 16). This is again consistent with the
results in Figure 14 that the maximum of OPE appears at a $NO_x$ level around 1 ppbv.

## 4. Discussion and Conclusions

On average, ozone production $P(O_3)$, was about 20-30 ppbv $hr^{-1}$ in the morning and 5-10

ppbv $hr^{-1}$ in the afternoon during DISCOVER-AQ in Houston in 2013. The diurnal variation of
$P(O_3)$ shows a broad peak in the morning with significant $P(O_3)$ in the afternoon obtained on ten
flight days in September 2013. High $P(O_3)$ mainly occurred with $L_N/Q$ greater than 0.5, i.e., in
the VOC sensitive regime. Since $P(O_3)$ depends on $NO_x$ levels and radical production rate, it
increases as [NO] increases up to ~1 ppbv and then levels off with further increases of [NO]. At
a given [NO], a higher production rate of $HO_x$ results in a higher ozone production rate. This has
implications for the $NO_x$ control strategies in order to achieve the ozone control goal.

The DISCOVER-AQ campaign in Houston is unique because of its large spatial coverage

and thus spatial variations of ozone production and its sensitivity to NOx and VOCs. Diurnal
variations of $P(O_3)$ at eight individual locations where the P-3B conducted vertical spirals show





that the $P(O_3)$ is on average more than 10 ppbv $hr^{-1}$ at locations with high $NO_x$ and VOC
emissions, such as Deer Park, Moody Tower, and Channelview, while at locations away from the
urban center with lower emissions of ozone precursors such as Galveston, Smith Point, and
Conroe, the ozone production rate is usually less than 10 ppbv $hr^{-1}$ on average. Hotspots of $P(O_3)$
were observed over Downtown Houston and Houston Ship Channel due to significant emissions
in these areas.
Ozone production tended more towards VOC sensitive in the morning with high $P(O_3)$
and in general, $NO_x$ sensitive in the afternoon with some exceptions. It was found that during
some afternoon time periods and locations, $P(O_3)$ was VOC sensitive. The diurnal variation of
$L_N/Q$ indicates that $P(O_3)$ was mainly VOC sensitive in the early morning and then transited
towards the $NO_x$ sensitive regime later in the day. High $P(O_3)$ in the morning was mainly
associated with VOC sensitivity due to high $NO_x$ levels in the morning. Specifically, Deer Park
was mostly VOC sensitive for the entire day, Moody Tower and Channelview were VOC
sensitive or in the transition regime, and Smith Point and Conroe were mostly $NO_x$ sensitive for
the entire day.
Based on the measurements on the P-3B, ozone production efficiency (OPE) was about 8
during DISCOVER-AQ 2013 in Houston. This OPE value is greater than the average OPE value
(5.9±1.2) obtained during the Texas Air Quality Study in 2006 (TexAQS2006), likely due to the
reduction in $NO_x$ emissions in Houston between 2006 and 2013 that pushed $NO_x$ levels closer to
1 ppbv in 2013 from higher NOx levels in previous years. This OPE value is about a factor of 1.5
to 2 higher than the OPE obtained in the DISCOVER-AQ 2011 study in Maryland due to higher
photochemical reactivity in Houston.
The results from this work strengthen our understanding of $O_3$ production; they indicate
that controlling $NO_x$ emissions will provide air quality benefits over the greater Houston
metropolitan area in the long run, but in selected areas controlling VOC emissions will also be
beneficial.

**Acknowledgements**
The authors acknowledge the entire DISCOVER-AQ science team for the use of the P-
3B measurement data in this work as well as Winston Luke and Paul Kelley at NOAA Air
Resources Laboratory for helpful discussion. This work was funded by the Texas Commission



on Environmental Quality (TCEQ) through the Air Quality Research Program (AQRP) at
University of Texas Austin (Contract #14-020). The contents, findings, opinions, and
conclusions are the work of the authors and do not necessarily represent the findings, opinions,
or conclusions of the TCEQ or AQRP. AQAST supported RRD.

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





**Table** 1. WRF and CMAQ model options that were used in both the original and improved
modeling scenarios.

| Weather Research and Forecasting (WRF) Version 3.6.1 Model Options | |
| --- | --- |
| Radiation | Long Wave: Rapid Radiative Transfer Model (RRTM) Short Wave: Goddard |
| Surface Layer | Pleim-Xiu |
| Land Surface Model | Pleim-Xiu |
| Boundary Layer | Asymmetric Convective Model (ACM2) |
| Cumulus | Kain-Fritsch |
| Microphysics | WRF Single-Moment 6 (WSM-6) |
| Nudging | Observational and analysis nudging |
| Damping | Vertical velocity and gravity waves damped at top of modeling domain |
| SSTs | Multi-scale Ultra-high Resolution (MUR) SST analysis (~1 km resolution) |
| Meteorological Initial and Boundary Conditions and Analysis Nudging Inputs | NAM 12 km |
| Observational Nudging Inputs | NCEP ADP Global Surface and Upper Air Observational Weather Data |
| **CMAQ Version 5.0.2 Model Options** | |
| Chemical Mechanism | Carbon Bond (CB05) |
| Aerosol Module | Aerosols with aqueous extensions version 5 (AE5) |
| Dry deposition | M3DRY |
| Vertical diffusion | Asymmetric Convective Model 2 (ACM2) |
| Emissions | 2012 TCEQ anthropogenic emissions Biogenic Emission Inventory System (BEIS) calculated within CMAQ |
| Chemical Initial and Boundary Conditions | Model for OZone and Related chemical Tracers (MOZART) Chemical Transport Model (CTM) |












**Figures:**

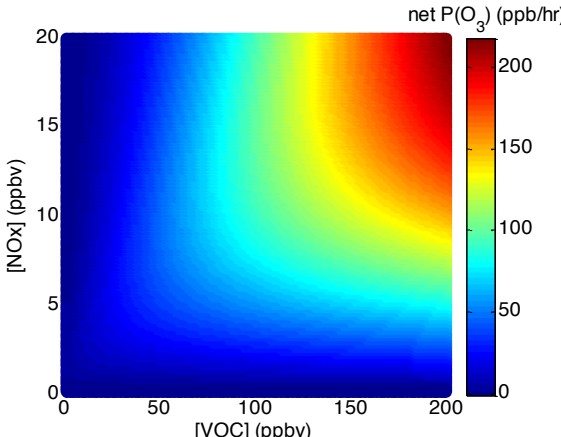


**Figure 1.** Ozone production empirical kinetic modeling approach (EKMA) diagram using a box
model results with NOx levels varying from 0-20 ppbv and VOC levels from 0-200 ppbv while
the mean concentrations of other species and the speciation of NOx and VOCs observed during
DISCOVER-AQ in Houston in 2013 were used to constrain the box model. This diagram clearly
shows the sensitivity of ozone production to $NO_x$ and VOCs in Houston.

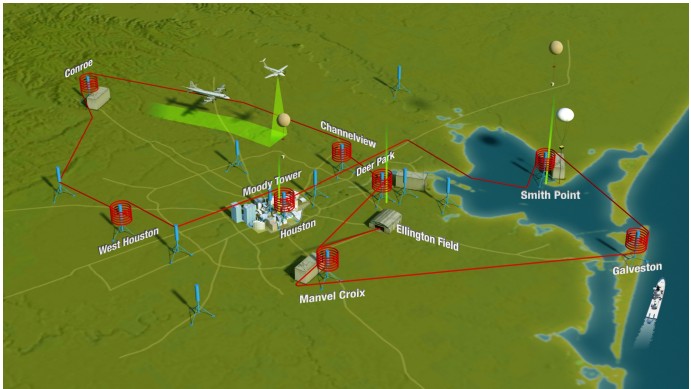


**Figure 2.** DISCOVER-AQ ground and spiral sites during the September 2013 Houston
campaign (http://discover-aq.larc.nasa.gov).





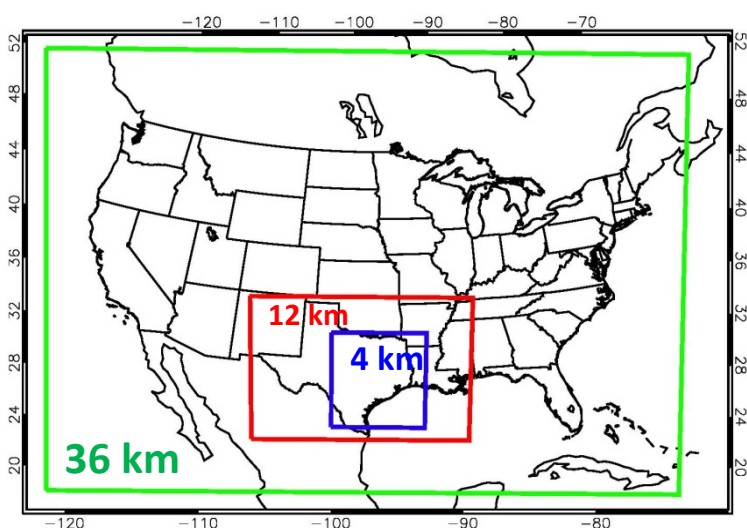


**Figure 3.** 36, 12, and 4 km CMAQ modeling domains

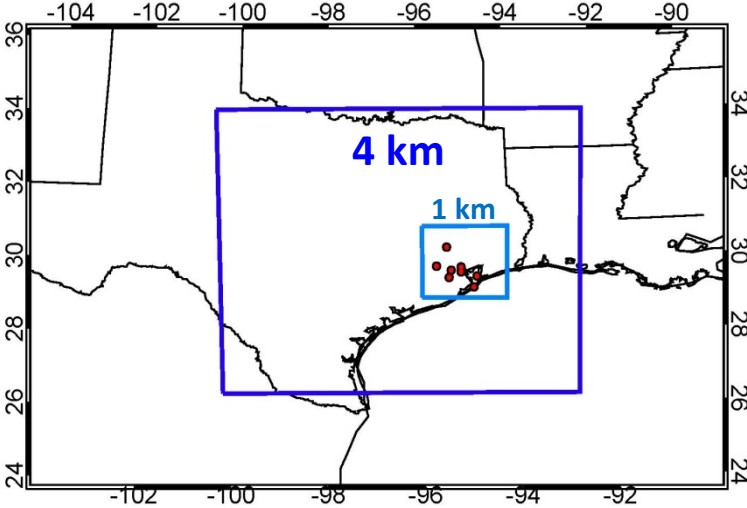


**Figure 4.** 4 and 1 km CMAQ modeling domains. The red dots show the NASA P-3B aircraft
spiral locations.







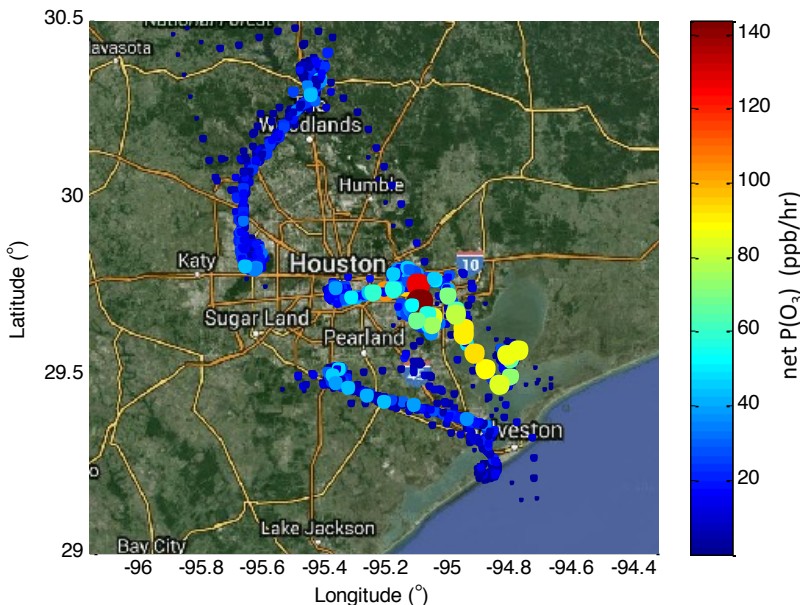


**Figure 5.** Net ozone production rate, net P(O₃) calculated using the box model results along the

P-3B flight track during DISCOVER-AQ in Houston in 2013. The size of dots is proportional to

P(O₃).



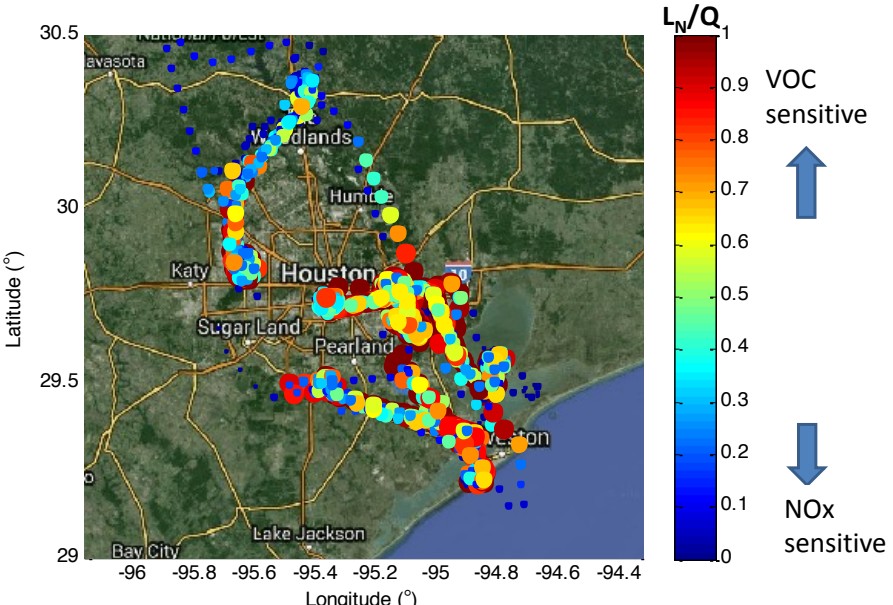


**Figure 6.** Ozone production sensitivity indicator, $L_N/Q$, along the P-3B flight track during DISCOVER-AQ in Houston in 2013. $P(O_3)$ is VOC-sensitive when $L_N/Q > 0.5$, and NOx-sensitive when $L_N/Q < 0.5$.

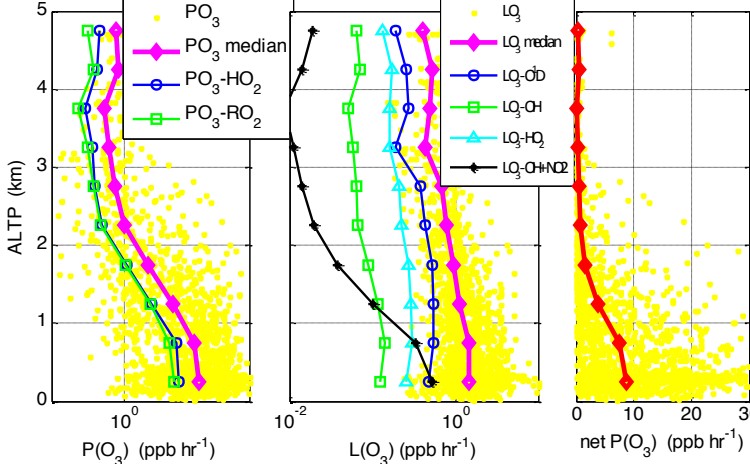


**Figure 7.** Vertical profiles of ozone production rate (left), ozone loss rate (middle), and net ozone production rate (right) during DISCOVER-AQ in Houston in 2013.




**Figure 8.** Diurnal variation of ozone production rate colored with the indicator $L_N/Q$ on ten

flight days during DISCOVER-AQ in Houston in 2013. The solid red circles represent the

median values in hourly bins of $P(O_3)$. Data are limited with the pressure altitude less than 1000

m to represent the lowest layer of the atmosphere.

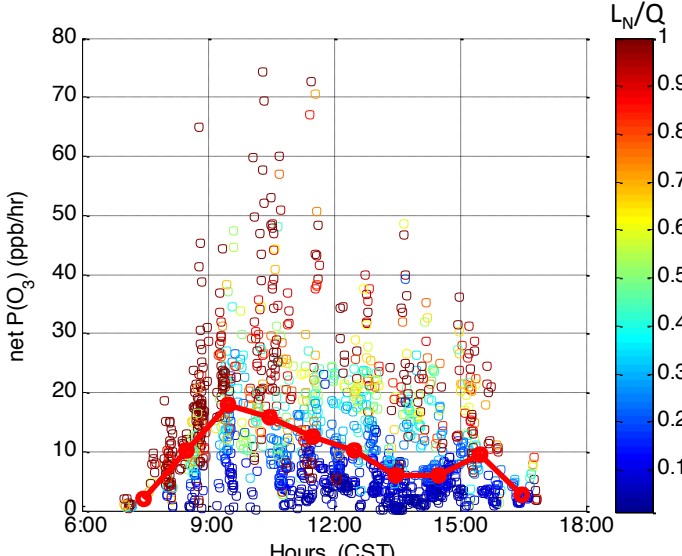

482

**Figure 9.** Diurnal variations of the indicator $L_N/Q$ of ozone production rate sensitivity colored

with ozone production rate (left) and NO and $NO_2$ concentrations (right) below 1000 m during

DISCOVER-AQ in Houston in 2013. The solid red circles are the median values in hourly bins

of $L_N/Q$.



487

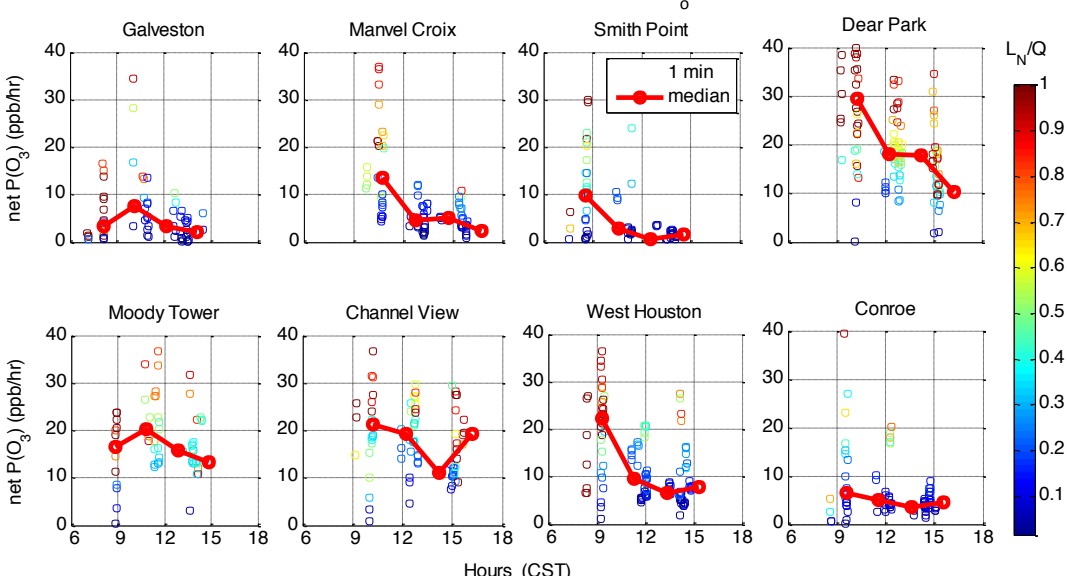

488

**Figure 10.** Diurnal variations of ozone production rate at eight individual spiral locations. Individual points are 1-min data colored with $L_N/Q$ and the linked red circles represent the median values in hourly bins of $P(O_3)$. Data are limited with the pressure altitude less than 1000 m to represent the lowest layer of the atmosphere.







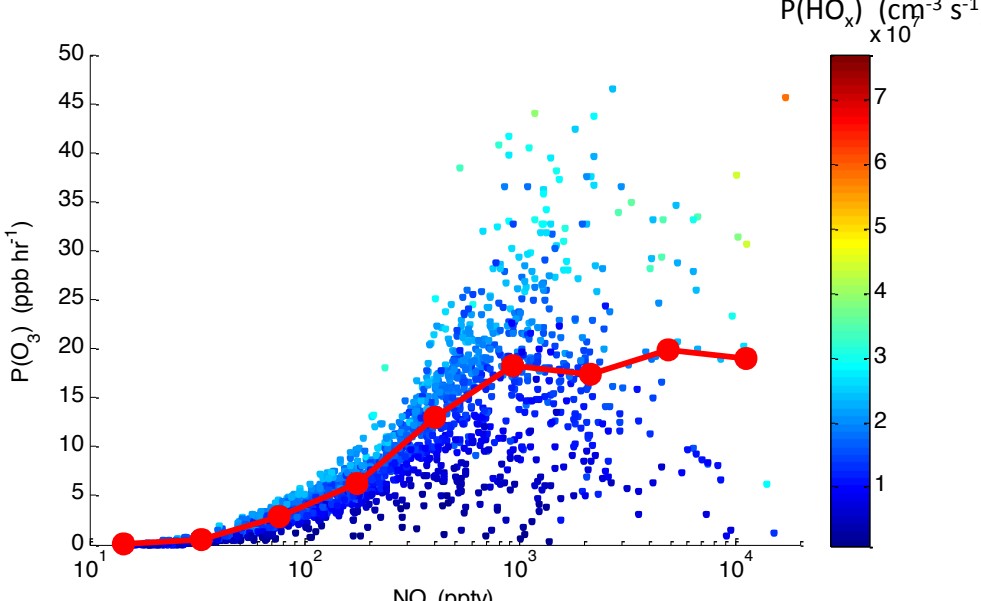


**Figure 11.** Ozone production as a function of NO mixing ratio. Individual data points are the 1-minute averages and are colored with the production rate of HOx (= OH + HO$_2$) during DISCOVER-AQ in Houston in 2013. The linked solid red circles represent the median values in [NO] bins. Note a log scale is used for the x-axis.






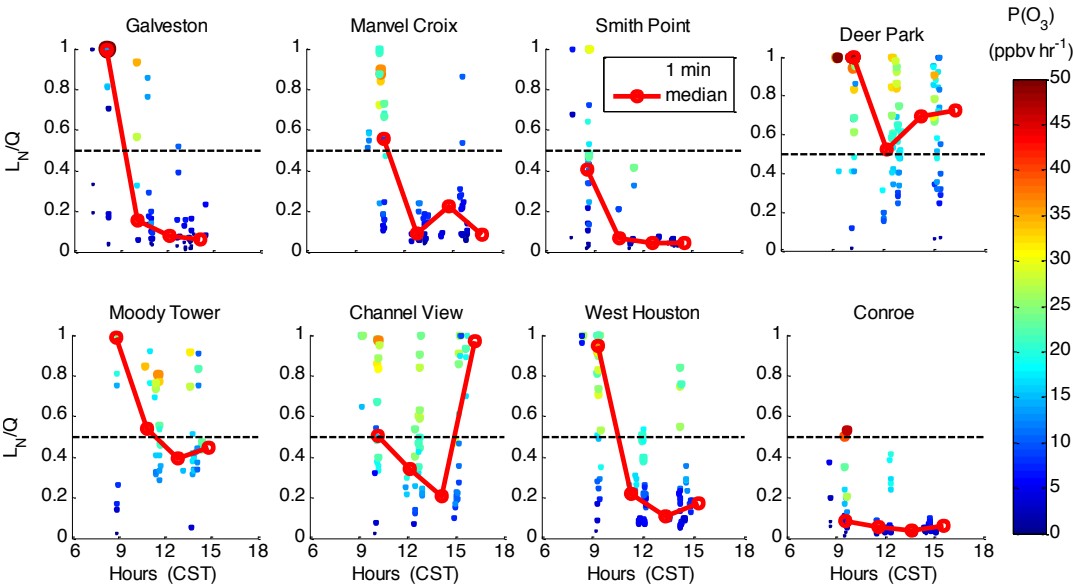


**Figure 12.** Diurnal variations of the indicator of ozone production sensitivity to NOx and VOCs, $L_N/Q$, at eight individual spiral locations during DISCOVER-AQ in Houston in 2013. Individual points are 1-min data colored by $P(O_3)$ and the linked red circles represent the median values in hourly bins of $P(O_3)$. Data are limited with the pressure altitude less than 1000 m to represent the lowest layer of the atmosphere.





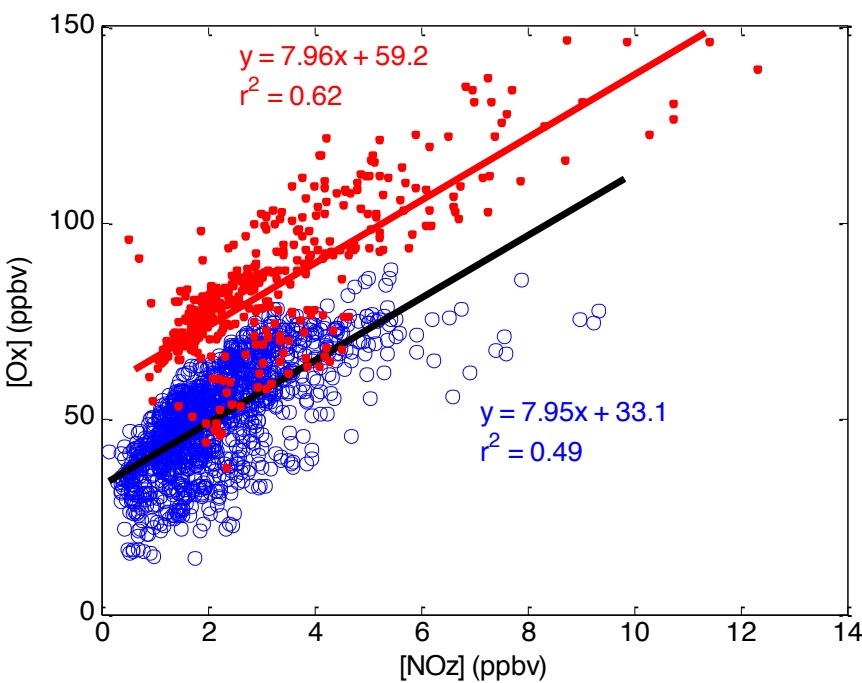

508

**Figure 13.** Photochemical oxidant, Ox (=$O_3$+$NO_2$) as a function of NOz (=NOy-NOx) during

DISCOVER-AQ in Houston in 2013. Red dots are the data collected on September 25 and 26,

2013 when high ambient ozone concentrations were observed. Blue circles are the data collected

during other flights. Data are limited with the pressure altitude less than 1000 m to represent the

lowest layer of the atmosphere.










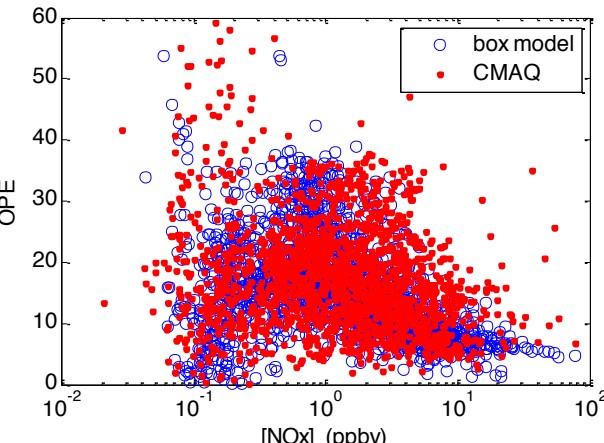


**Figure 14.** Ozone production efficiency (OPE) versus NOx in the box model (blue circles) and
CMAQ model (red dots) results. OPE is calculated according to its definition as the net ozone
formation rate divided by of the formation rate of NOz.


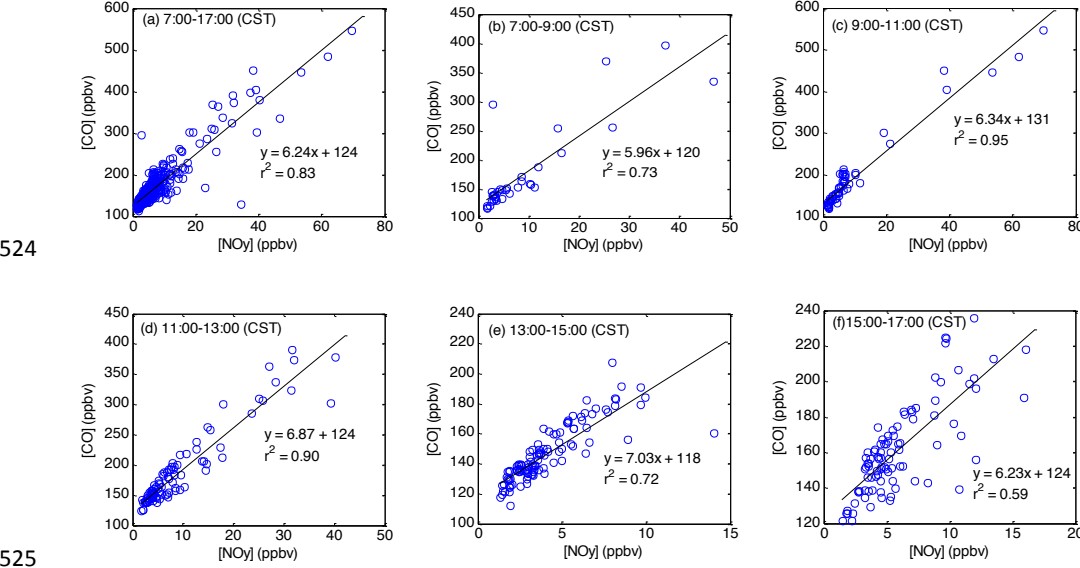



**Figure 15.** CO versus NOy and linear regression on September 25 and 26 at different times of
the day: (a) 07:00-17:00 (all data), (b) 07:00-09:00, (c) 09:00-11:00, (d) 11:00-13:00, (e) 13:00-
15:00, and (f) 15:00-17:00 (CST).




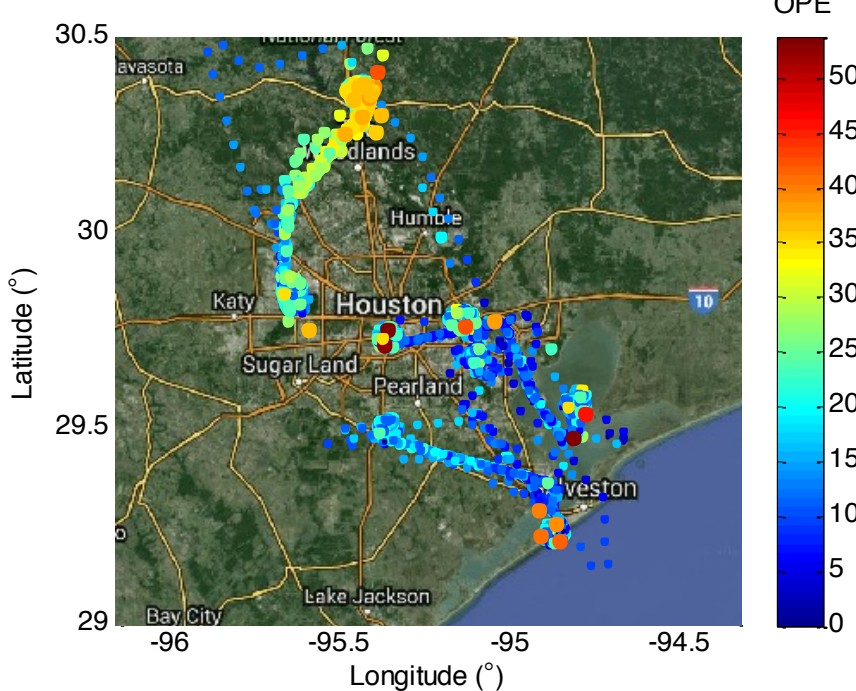


**Figure 16.** Ozone production efficiency (OPE) along the P-3B flight track during DISCOVER-
AQ in Houston in 2013. OPE was calculated using the box model results as the ratio of net ozone
formation rate to the formation rate of NOz.
