# Peer review of "Ozone Production and Its Sensitivity to NOx and VOCs: Results from the DISCOVER-AQ"

_Atmospheric Chemistry and Physics, 2016_

## Referee Comment (RC1) · Anonymous Referee #1 · 27 May 2016

Review of "Ozone production and its sensitivity to NOX and VOCs: results from the DISCOVER-AQ field experiment, Houston 2013"

Mazzuca et al

The authors estimate ozone production efficiency using aircraft and surface measurements from the 2013 Houston DISCOVER-AQ field campaign. The DISCOVER-AQ field campaigns were designed to provide surface and sub-orbital measurements for validation of satellite air quality products. In addition to using ambient data as input to a photochemical box model, a 3D photochemical grid model was also used to estimate ozone production efficiency. The authors find that OPE is lower than in the numerous previous Texas field campaigns and somewhat similar to the Baltimore DISCOVER-AQ

field study. The difference between earlier Texas field studies was attributed to lower NOX and VOC emissions in the Houston area due to emissions control plans.

Overall the information about ozone production efficiency is well presented and well articulated by the authors. This work does not present a lot of new information about the Houston area or present any clear implications about emissions control plans. Should Texas implement morning VOC controls and area wide day-long NOX controls to decrease ozone production in the area? The authors state several times that these results have important emissions control policy implications but it is not clear what type of program implementation would be needed based on the diurnal ozone production efficiencies presented here.

The estimates of O3 production efficiency and comparison with previous Houston field experiments and the Baltimore campaign are the most interesting aspects of the study. Given that this paper is focused on NOX and VOC contribution to O3 production the authors should provide NOX and VOC measurements from this study and also compare those with previous Houston field studies to provide more context about how these pollutants are decreasing and for VOC how total VOC and VOC reactivity is decreasing to support conclusions about ozone production efficiency. Also, a comparison with another area like Baltimore would be useful.

The authors provide CMAQ simulated ozone production efficiency but provide no information about the emission inventory used for the simulation and how well the model predicted NOX, NOZ, VOC, and O3 compared with the aircraft and surface measurements made during the field study. Is it ok that the model predicts a similar OPE to the box model but not capture the magnitudes of the precursors or ozone correctly? The information presented about OPE is useful, but additional work is needed for this to provide a more comprehensive understanding of ozone production in Houston with respect to the models used by regulators for decision support and context from the many previous Houston field studies.

[Figure]

Specific comments:

The last half of the introduction section reads like a white paper on the Houston DISCOVER-AQ field study. Since this paper does not present any information relevant to the mission of that field study which was to validate satellite measurements the discussion of the DISCOVER-AQ campaign could be de-emphasized in favor of more time spent on the multitude of historical field studies in the Houston area. Also, the authors never clearly state in the introduction what they are presenting and why that information is novel.

The authors do not need to explain why CB05 is used rather than CBIV, but an explanation about why CB05 was used rather than the newer version CB6 is necessary. At several points in the manuscript the authors note than organic nitrate fate can confound OPE interpretation so the choice of an older Carbon Bond mechanism that has a less realistic treatment of organic nitrates is needed. Also, it is not clear why all species have the same 2 day deposition lifetime. Species like O3 and HNO3 deposit out of the atmosphere and very different rates.

Please provide information about the emission inventory and modeling used as input to the CMAQ simulation and the source of the initial and boundary conditions.

In the results section, please provide some comparison of CMAQ estimated VOC, speciated VOC, NO, NO2, HNO3, PANs, HNO3, and O3 with measurements.

The authors suggest one difference in OPE between Houston and Baltimore is due to reactivity. Please provide speciated VOC concentrations from each field study by reactivity so this relationship is clearer.

The authors make a lot of strong conclusions about trends in OPE when NOX is greater or less than 1 ppb as shown in Figure 14. The points in Figure 14 do not show a distinct relationship above or below any level of the NOX concentrations. Perhaps box plots binned by NOX concentration would be a better way to show this type of relationship (if

it really exists).
* * *

---

## Referee Comment (RC2) · Anonymous Referee #2 · 3 Jun 2016

This paper by Mazzuca et al. presents modeling and data analysis results aimed at characterizing ozone production rates and ozone production efficiency in various locations around Houston, TX, during the DISCOVER-AQ campaign of September 2013. In general, the paper is well written, uses mostly adequate citations, and has an appropriate abstract. However, I believe that some of the figures are not necessary and that some of them provide very little new insight. The analyses performed and the approach used are tried and true so technically, there are no major faults with the work (though I question the use of a box model in Houston when the meteorology is so complex - why not just use the 3D model as it can provide answers to some of the questions asked and the ambient data can be used for model evaluation). However, due to a lack of

novelty and a lack of truly new findings that warrant an entire manuscript, I am unable to recommend this manuscript for publication in ACP.

With regard to figures, Figure 1 is not necessary (the ozone isopleth is "classic"), Figure 2 would be better as a map with points/labels as the extraneous stuff is distracting, and Figures 3 and 4 can be combined. In addition, some of the figures are intuitive based on previous work in Houston and other locations (5, 6, 8, and 9).

My largest criticism of this work is that it is known from three previous field campaigns that ozone production rates and sensitivities in Houston are temporally and spatially dependent. It seems to be that the most new information appears on lines 203-205 (line 206 is intuitive) regarding O3 loss and the split between RO2 and HO2 reactions with NO (unless this information is published elsewhere and I am unaware) and on line 255+ where it is noted that OPE has decreased in Houston compared to previous campaigns (due to the decrease in NOx emissions). I do not believe that these warrant a manuscript by themselves.

The authors do not put Houston in the context of other locations. For example, they state on line 68 that "there are a limited number of observation-based studies on ozone production and its sensitivity to NOx and VOCs." There have been such studies made in Houston (SHARP, TEXAQS I and II) as well as in other locations across the US (Nashville, New England) and Europe. It would be appropriate to make such comparisons.

A minor comment on the box model. What is the basis for assuming a two-day lifetime for all calculated species to avoid build up? Could a citation be provided? Or could other meteorological models be used to provide a more accurate estimate (this gets back to use of a 3D versus box model)?

———————————————————

---

## Author Comment (AC1) · 27 Jul 2016

Response to Anonymous Referee #1: 02 June 2016

We thank the reviewer for providing insightful comments and helpful suggestions that have substantially improved the manuscript. Below we have included the review comments followed by our responses in italic. In the revision of this manuscript, we have highlighted those changes accordingly with track change.

1) Review of "Ozone production and its sensitivity to NOX and VOCs: results from the DISCOVER-AQ field experiment, Houston 2013" The authors state several times that these results have important emissions control policy implications but it is not clear
what type of program implementation would be needed based on the diurnal ozone production efficiencies presented here.

Response: We are not suggesting a specific implementation program (which is beyond the scope of this work), however, are suggesting that it may be more beneficial at certain locations, during certain times of day, to regulate VOCs based on the diurnal ozone production efficiencies we report. We are providing a scientific basis through which policy makers could develop an emission reduction strategy.

2) Given that this paper is focused on NOX and VOC contribution to O3 production the authors should provide NOX and VOC measurements from this study and also compare those with previous Houston field studies to provide more context about how these pollutants are decreasing and for VOC how total VOC and VOC reactivity is decreasing to support conclusions about ozone production efficiency. Also, a comparison with another area like Baltimore would be useful.

Response: Both NOx and VOC levels in Houston have been continuously decreasing in the past 15-20 years as shown in Figure 1(S1 in paper), the time series of NOx, ethane, and propene at two monitoring sites near the Houston Ship Channel.

Figure 1 caption: Time series of NO, NOx, ethane and propene concentrations at the Deer Park and Clinton sites from 1998 to 2014. The Deer Park site is located in southeast of the Ship Channel. The Clinton site is located on the northwestern end of the Ship Channel. Each data point represents an average of hourly samples collected between July 1 and November 30 for each year. Missing data points indicate that too few valid samples (< 70%) were collected during that year. NO and NOx* data collected hourly using chemiluminescence sampler with molybdenum catalyst to convert NOx* (not true NOx because Mo catalyst converts other N species besides NO2 to NO) to NO. VOC data collected over a 40-minute period each hour using automated gas chromatography with cryogenic pre-concentration.

The NOx levels and OH reactivity in Houston during DAQ2013 and in Maryland during

[Figure]

DAQ2011 are quite different, as shown in Figure 2. Houston has much higher NOx levels throughout the day. For OH reactivity, it is greater in Houston than in Maryland in the morning, but is comparable in both locations in the afternoon. Note as shown in Figure 4, due to different emission sources, in Houston anthropogenic VOCs are the main contributor to the OH reactivity from VOCs, while in Maryland, biogenic VOCs (mainly isoprene) dominates the OH reactivity from VOCs. Different NOx levels and different VOC sources in Houston and Maryland are responsible for the different OPE values in the two areas.

Figure 2 caption: Diurnal variations of NOx (left) and OH reactivity (Right) in Houston (linked blue circles) during DAQ2013 and in Maryland (linked red triangles) during DAQ2011.

3) The authors provide CMAQ simulated ozone production efficiency but provide no information about the emission inventory used for the simulation and how well the model predicted NOX, NOZ, VOC, and O3 compared with the aircraft and surface measurements made during the field study. Is it ok that the model predicts a similar OPE to the box model but not capture the magnitudes of the precursors or ozone correctly? The information presented about OPE is useful, but additional work is needed for this to provide a more comprehensive understanding of ozone production in Houston with respect to the models used by regulators for decision support and context from the many previous Houston field studies.

Response: The WRF and CMAQ model options are described in Table 1. In Section 2.3, we also added the following a few sentences to describe the emissions we used in the CMAQ simulations: "The 2012 baseline anthropogenic emissions from the Texas Commission on Environmental Quality (TCEQ) were used as input to CMAQ. These emissions contain the most-up-to-date Texas anthropogenic emissions inventory and a compilation of emissions estimates from Regional Planning Offices throughout the US. Biogenic emissions were calculated online within CMAQ with Biogenic Emission Inventory System (BEIS). Lightning emissions were also calculated online within CMAQ."

CMAQ simulated a high bias in surface and aloft ozone (Tables 1). CMAQ also simulated a low bias in CO, CH2O, isoprene, NO2, and NO aloft and a high bias in NOy aloft (Table 2). Recent work has shown that oceanic emissions of iodine and bromine result in ozone destruction (Carpenter et al., 2013). The high ozone bias in our results is expected due to the lack of oceanic iodine and bromine emissions and the associated chemistry. Biases in surface ozone are larger near the coastline (i.e., Galveston) than sites inland (i.e., Conroe).

Table 1 caption. Mean bias (MB), normalized mean bias (NMB), normalized mean error (NME), root mean square error (RMSE), and Gross Error (GE) of surface ozone for the 2nd iterative 1 km WRF simulations covering all of September 2013.

Table 2 caption. Second iterative 1 km CMAQ simulated mean bias (MB), normalized mean bias (NMB), normalized mean error (NME), and root mean square error (RMSE) of O3, CO, CH2O, Isoprene (ISO), NO2, NO, and NOy covering measurements made onboard the NASA P-3B aircraft on all flight days during the DISCOVER-AQ field campaign.

4) The last half of the introduction section reads like a white paper on the Houston DISCOVER-AQ field study. Since this paper does not present any information relevant to the mission of that field study which was to validate satellite measurements the discussion of the DISCOVER-AQ campaign could be de-emphasized in favor of more time spent on the multitude of historical field studies in the Houston area. Also, the authors never clearly state in the introduction what they are presenting and why that information is novel.

Response: We have removed lines 89-96 and combine lines 97 – 100 and took out lines 102-106. We edited lines 81-84 to read: "In the work presented here, we provide investigations of spatial and temporal variations of ozone production and its sensitivity to NOx and VOCs to provide a scientific basis to develop a non-uniform emission reduction strategy for O3 pollution control in urban areas such as Houston."

5) The authors do not need to explain why CB05 is used rather than CBIV, but an explanation about why CB05 was used rather than the newer version CB6 is necessary. At several points in the manuscript the authors note than organic nitrate fate can confound OPE interpretation so the choice of an older Carbon Bond mechanism that has a less realistic treatment of organic nitrates is needed. Also, it is not clear why all species have the same two-day deposition lifetime. Species like O3 and HNO3 deposit out of the atmosphere and very different rates.

Response: CB05 is the most up to date Carbon Bond mechanism in CMAQ (i.e., CB6 has not been implemented into CMAQ at the time the analysis was performed). The box model was constrained for all long-lived measured species like ozone and HNO3 and we do not assume a two-day deposition lifetime. An additional two-day lifetime due to deposition and heterogeneous losses is assumed for calculated species in the box model. Most calculated species like OH, HO2 and RO2 are reactive intermediates and have lifetimes on the order of seconds to minutes, much shorter than 2 days. Adding this additional two-day lifetime would not affect the model results at all. There are a few long-lived species (like organic acid and alcohols) calculated in the model that could potentially accumulate to levels much higher than the levels in the ambient air. We have revised this sentence: "An additional lifetime of two days was assumed for some calculated long lived species such as organic acids and alcohols to avoid unexpected accumulation of these species in the model."

6) Please provide information about the emission inventory and modeling used as input to the CMAQ simulation and the source of the initial and boundary conditions.

Response: The WRF and CMAQ model options have been described in Table 1. In Section 2.3, we also added the following a few sentences for the emissions we used in the CMAQ simulations: "The 2012 baseline anthropogenic emissions from the Texas Commission on Environmental Quality (TCEQ) were used as input to CMAQ. These emissions contain the most-up-to-date Texas anthropogenic emissions inventory and a compilation of emissions estimates from Regional Planning Offices throughout the

US Biogenic emissions was calculated online within CMAQ with Biogenic Emission Inventory System (BEIS). Lightning emissions were also calculated online within CMAQ." It is also listed in Table 1 of this manuscript.

7) In the results section, please provide some comparison of CMAQ estimated VOC, speciated VOC, NO, NO2, HNO3, PANs, HNO3, and O3 with measurements.

Response: An evaluation of the improved WRF and CMAQ model simulations for the entire month of September 2013 was conducted. Statistics used to evaluate WRF and CMAQ are described Tables 3. CMAQ simulated a high bias in surface and aloft ozone (Tables 1). CMAQ also simulated a low bias in CO, CH2O, isoprene, NO2, and NO aloft and a high bias in NOy aloft (Table 2). Recent work has shown that oceanic emissions of iodine and bromine result in ozone destruction. The high ozone bias in our results is expected due to the lack of oceanic iodine and bromine emissions and the associated chemistry. Biases in surface ozone are larger near the coastline (i.e., Galveston) than sites inland (i.e., Conroe) as shown in Figure 7-3.

Table 3 caption. Definition of the statistics used in WRF and CMAQ model evaluations. In these equations M represents the model results, O represents the observations, and N is the number of data points.

Table 4 caption. Mean bias (MB), normalized mean bias (NMB), normalized mean error (NME), root mean square error (RMSE), and Gross Error (GE) of 2 m temperature, 10 m wind speed, and 10 m wind direction for the 2nd iterative 1 km WRF simulations covering all of September 2013.

Figure 3 caption. Observed (*) and CMAQ simulated (solid lines) maximum 8 hour average ozone at La Porte Sylvan Beach (red), Conroe (purple), Galveston (blue), and West Houston (green) during September 2013.

8) The authors suggest one difference in OPE between Houston and Baltimore is due to reactivity. Please provide speciated VOC concentrations from each field study by

reactivity so this relationship is clearer.

Response: The median OH reactivity due to non-methane hydrocarbons (NMHCs) was 3.3 s-1 observed during DISCOVER-AQ 2013 in Houston and 1.2 s-1 observed during DISCOVER-AQ 2011 in Maryland. As shown in Figure 2, alkanes and alkenes were dominant contributors to the OH reactivity due to NMHCs in Houston in 2013, while isoprene and alkanes were dominant contributors to the OH reactivity due to NMHCs in Maryland in 2011. The differences in overall OH reactivity and its distributions in the two locations are responsible to the different OPEs in the two different environments. We have included this in the Supporting Information.

Figure 4 caption. Distributions of OH reactivity due to non-methane hydrocarbons in DISCOVER-AQ 2011 in Maryland (left) and 2013 in Houston (right).

9) The authors make a lot of strong conclusions about trends in OPE when NOX is greater or less than 1 ppb as shown in Figure 14. The points in Figure 14 do not show a distinct relationship above or below any level of the NOX concentrations. Perhaps box plots binned by NOX concentration would be a better way to show this type of relationship (if it really exists).

Response: We have updated Figure 14 (now Figure 13) by adding median OPE values binned by NOx concentration on top of the individual data points and the trend seems more distinct.

Figure 5 (13) caption. Ozone production efficiency (OPE) versus NOx in the box model (blue circles) and the CMAQ model pink dots) results. The linked blue circles show the median OPE values binned by NOx concentration in the box model, while the linked red triangles show the median OPE values binned by NOx concentration in the CMAQ model, OPE is calculated according to its definition as the net ozone formation rate divided by of the formation rate of NOz.

Please also note the supplement to this comment:

[Figure]

http://www.atmos-chem-phys-discuss.net/acp-2016-215/acp-2016-215-AC1-supplement.pdf

[Figure]

[Figure]

**Fig. 1.** Fig1:Time series of NO, NOx, ethane and propene concentrations at the Deer Park and Clinton sites from 1998 to 2014.

[Figure]

[Figure]

**Fig. 2.** Fig2:Diurnal variations of NOx (left) and OH reactivity (Right) in Houston (linked blue circles) during DAQ2013 and in Maryland (linked red triangles) during DAQ2011.

|  | Surface Ozone (ppbv) |
|---|---|
| MB | 9.5 |
| NMB (%) | 39 |
| NME (%) | 51 |
| RMSE | 15 |
| GE | 12 |

**Fig. 3.** Table 1. Mean bias (MB), normalized mean bias (NMB), normalized mean error (NME), root mean square error (RMSE), and Gross Error (GE) of surface ozone

|  | | O$_3$ | CO | CH$_2$O | ISO | NO2 | NO | NOy |
|---|---|---|---|---|---|---|---|---|
| Model | MB | 0.8 | -5.8 | -0.3 | -0.02 | -0.5 | -0.3 | 0.04 |
| | NMB | 1.4 | -4.8 | -16 | -7.7 | -39 | -66 | 1.3 |
| | NME | 15 | 17 | 37 | 70 | 70 | 84 | 61 |
| | RMSE | 12 | 35 | 1.4 | 0.7 | 3.1 | 2.2 | 4.7 |

**Fig. 4.** Table 2. Second iterative 1 km CMAQ simulated mean bias (MB), normalized mean bias (NMB), normalized mean error (NME), and root mean square error (RMSE) of O3, CO, CH2O, Isoprene (ISO), NO2, NO, etc.

| Mean Bias (MB) | $MB = \dfrac{1}{N}\sum_{i=1}^{N}(M_i - O_i)$ |
|---|---|
| Normalized Mean Bias (NMB) | $NMB = \dfrac{\sum_{i=1}^{N}(M_i - O_i)}{\sum_{i=1}^{N} O_i} \times 100\%$ |
| Normalized Mean Error (NME) | $NME = \dfrac{\sum_{i=1}^{N}|M_i - O_i|}{\sum_{i=1}^{N} O_i} \times 100\%$ |
| Root Mean-Square Error (RMSE) | $RMSE = \sqrt{\dfrac{1}{N}\sum_{i=1}^{N}(M_i - O_i)^2}$ |
| Gross Error (G) | $GE = \dfrac{1}{N}\sum_{i=1}^{N}|M_i - O_i|$ |

**Fig. 5.** Table 3. Definition of the statistics used in WRF and CMAQ model evaluations. In these equations M represents the model results, O represents the observations, and N is the number of data points.

[Figure]

| | 2 m Temperature (K) | | 10 m Wind Speed (m/s) | | 10 m Wind Direction (deg) | |
|---|---|---|---|---|---|---|
| | | Model | | Model | | Model |
| MB | | 0.2 | | -0.8 | | 32 |
| NMB (%) | | 0.1 | | -17 | | 26 |
| NME (%) | | 0.4 | | 36 | | 26 |
| RMSE | | 1.6 | | 2.3 | | 51 |
| GE | | 1.2 | | 1.7 | | 32 |

**Fig. 6.** Table 4. Mean bias (MB), normalized mean bias (NMB), normalized mean error (NME), root mean square error (RMSE), and Gross Error (GE) of 2 m temperature, 10 m wind speed, and 10 m wind direction

[Figure]

**Fig. 7.** Fig3:Observed (*) and CMAQ simulated (solid lines) maximum 8 hour average ozone at La Porte Sylvan Beach (red), Conroe (purple), Galveston (blue), and West Houston (green) during September 2013.

[Figure]

**Fig. 8.** Fig4:Distributions of OH reactivity due to non-methane hydrocarbons in DISCOVER-AQ
2011 in Maryland (left) and 2013 in Houston (right).

[Figure]

**Fig. 9.** Fig5:Ozone production efficiency (OPE) versus NOx in the box model (blue circles) and the CMAQ model pink dots) results.

**Supplement:**

Supporting Information for:

**Ozone Production and Its Sensitivity to NO$_x$ and VOCs: Results from the DISCOVER-AQ Field Experiment, Houston 2013**

Gina M. Mazzuca[1], Xinrong Ren[1,2,*], Christopher P. Loughner[2,3,4], Mark Estes[5], James H. Crawford[6], Kenneth E. Pickering[1,4], Andrew J. Weinheimer[7], and Russell R. Dickerson[1]

[1]Department of Atmospheric and Oceanic Science, University of Maryland, College Park, MD 20742, USA

[2]Air Resources Laboratory, National Oceanic and Atmospheric Administration, College Park, MD 20740, USA

[3]Earth System Science Interdisciplinary Center, University of Maryland, College Park, MD 20740, USA

[4]NASA Goddard Space Flight Center, Greenbelt, MD 20771, USA

[5]Texas Commission on Environmental Quality, Austin, TX 78711, USA

[6]NASA Langley Research Center, Hampton, VA 23681, USA

[7] National Center for Atmospheric Research, Boulder, CO 80307, USA

*Correspondence to: X. Ren (ren@umd.edu)

SI 1.

Both NOx and VOC levels in Houston have been continuously decreasing in the past 15-20 years as shown in Figure 1, the time series of NOx, ethane, and propene at two monitoring sites near the Houston Ship Channel.

[Figure]

[Figure]

**Figure S1.** Time series of NO, NOx, ethane and propene concentrations at the Deer Park and Clinton sites from 1998 to 2014. The Deer Park site is located in southeast of the Ship Channel. The Clinton site is located on the northwestern end of the Ship Channel. Each data point represents an average of hourly samples collected between July 1 and November 30 for each year. Missing data points indicate that too few valid samples (< 70%) were collected during that year. NO and NOx* data collected hourly using chemiluminescence sampler with molybdenum catalyst to convert NOx* (not true NOx because Mo catalyst converts other N species besides NO2 to NO) to NO. VOC data collected over a 40 minute period each hour using automated gas chromatography with cryogenic pre-concentration.

The NOx levels and OH reactivity in Houston during DAQ2013 and in Maryland during DAQ2011 are quite different, as shown in Figure 2. Houston has much higher NOx levels throughout the day. For OH reactivity, it is greater in Houston than in Maryland in the morning, but is comparable in both location in the afternoon. Note as shown in Figure 4, due to different emission sources, in Houston anthropogenic VOCs are the main contributor to the OH reactivity from VOCs, while in Maryland, biogenic VOCs (mainly isoprene) dominates the OH reactivity from VOCs. Different NOx levels and different VOC sources in Houston and Maryland are responsible for the different OPE values in the two areas.

[Figure]

**Figure S2.** Diurnal variations of NOx (left) and OH reactivity (Right) in Houston (linked blue circles) during DAQ2013 and in Maryland (linked red triangles) during DAQ2011.

SI 2.

CMAQ simulated a high bias in surface and aloft ozone (Table 1). CMAQ also simulated a low bias in CO, $CH_2O$, isoprene, $NO_2$, and NO aloft and a high bias in NOy aloft (Table 2). Recent work has shown that oceanic emissions of iodine and bromine result in ozone destruction (Carpenter et al., 2013). The high ozone bias in our results is expected due to the lack of oceanic iodine and bromine emissions and the associated chemistry. Biases in surface ozone are larger near the coastline (i.e., Galveston) than sites inland (i.e., Conroe).

**Table S1.** Mean bias (MB), normalized mean bias (NMB), normalized mean error (NME), root mean square error (RMSE), and Gross Error (GE) of surface ozone for the 2nd iterative 1 km WRF simulations covering all of September 2013.

|  | **Surface Ozone (ppbv)** |
| --- | --- |
| MB | 9.5 |
| NMB (%) | 39 |
| NME (%) | 51 |

| | |
|---|---|
| RMSE | 15 |
| GE | 12 |

**Table S2.** Second iterative 1 km CMAQ simulated mean bias (MB), normalized mean bias (NMB), normalized mean error (NME), and root mean square error (RMSE) of $O_3$, CO, $CH_2O$, Isoprene (ISO), $NO_2$, NO, and NOy covering measurements made onboard the NASA P-3B aircraft on all flight days during the DISCOVER-AQ field campaign

| | | $O_3$ | CO | $CH_2O$ | ISO | NO2 | NO | NOy |
|---|---|---|---|---|---|---|---|---|
| **Model** | MB | 0.8 | -5.8 | -0.3 | -0.02 | -0.5 | -0.3 | 0.04 |
| | NMB | 1.4 | -4.8 | -16 | -7.7 | -39 | -66 | 1.3 |
| | NME | 15 | 17 | 37 | 70 | 70 | 84 | 61 |
| | RMSE | 12 | 35 | 1.4 | 0.7 | 3.1 | 2.2 | 4.7 |

SI 3.

An evaluation of the improved WRF and CMAQ model simulations for the entire month of September 2013 was conducted. Statistics used to evaluate WRF and CMAQ are described Table S3. CMAQ simulated a high bias in surface and aloft ozone (Table S1). CMAQ also simulated a low bias in CO, $CH_2O$, isoprene, $NO_2$, and NO aloft and a high bias in NOy aloft (Table S2). Recent work has shown that oceanic emissions of iodine and bromine result in ozone destruction. The high ozone bias in our results is expected due to the lack of oceanic iodine and bromine emissions and the associated chemistry. Biases in surface ozone are larger near the coastline (i.e., Galveston) than sites inland (i.e., Conroe) as shown in Figure S3.

**Table S3.** Definition of the statistics used in WRF and CMAQ model evaluations. In these equations M represents the model results, O represents the observations, and N is the number of data points.

| | |
|---|---|
| Mean Bias (MB) | $$MB = \frac{1}{N}\sum_{i=1}^{N}(M_i - O_i)$$ |
| Normalized Mean Bias (NMB) | $$NMB = \frac{\sum_{i=1}^{N}(M_i - O_i)}{\sum_{i=1}^{N}O_i} \times 100\%$$ |
| Normalized Mean Error (NME) | $$NME = \frac{\sum_{i=1}^{N}|M_i - O_i|}{\sum_{i=1}^{N}O_i} \times 100\%$$ |
| Root Mean-Square Error (RMSE) | $$RMSE = \sqrt{\frac{1}{N}\sum_{i=1}^{N}(M_i - O_i)^2}$$ |

| Gross Error (G) | $GE = \dfrac{1}{N} \displaystyle\sum_{i=1}^{N} |M_i - O_i|$ |
|---|---|

**Table S4.** Mean bias (MB), normalized mean bias (NMB), normalized mean error (NME), root mean square error (RMSE), and Gross Error (GE) of 2 m temperature, 10 m wind speed, and 10 m wind direction for the 2[nd] iterative 1 km WRF simulations covering all of September 2013.

| | 2 m Temperature (K) | | 10 m Wind Speed (m/s) | | 10 m Wind Direction (deg) | |
|---|---|---|---|---|---|---|
| | | Model | | Model | | Model |
| MB | | 0.2 | | -0.8 | | 32 |
| NMB (%) | | 0.1 | | -17 | | 26 |
| NME (%) | | 0.4 | | 36 | | 26 |
| RMSE | | 1.6 | | 2.3 | | 51 |
| GE | | 1.2 | | 1.7 | | 32 |

[Figure]

**Figure S3.** Observed (*) and CMAQ simulated (solid lines) maximum 8 hour average ozone at La Porte Sylvan Beach (red), Conroe (purple), Galveston (blue), and West Houston (green) during September 2013.

SI 4.

The median OH reactivity due to non-methane hydrocarbons (NMHCs) was 3.3 s$^{-1}$ observed during DISCOVER-AQ 2013 in Houston and 1.2 s$^{-1}$ observed during DISCOVER-AQ 2011 in Maryland. As shown in Figure 2, alkanes and alkenes were dominant contributors to the OH reactivity due to NMHCs in Houston in 2013, while isoprene and alkanes were dominant contributors to the OH reactivity due to NMHCs in Maryland in 2011. The differences in overall OH reactivity and its distributions in the two locations are responsible to the different OPEs in the two different environments.

[Figure]

**Figure S4.** Distributions of OH reactivity due to non-methane hydrocarbons in DISCOVER-AQ 2011 in Maryland (left) and 2013 in Houston (right).

---

## Author Comment (AC2) · 27 Jul 2016

Response to Anonymous Referee #2: 10 July 2016

We thank the reviewer for providing insightful comments and helpful suggestions that have substantially improved the manuscript. Below we have included the review comments followed by our responses. In the revision of this manuscript, we have highlighted those changes accordingly in blue. Supplemental information is also provided.

1. The analyses performed and the approach used are tried and true so technically, there are no major faults with the work (though I question the use of a box model in Houston when the meteorology is so complex - why not just use the 3D model as it

can provide answers to some of the questions asked and the ambient data can be used for model evaluation). However, due to a lack of novelty and a lack of truly new findings that warrant an entire manuscript, I am unable to recommend this manuscript for publication in ACP.

Response: The reviewer initially states that the work is not recommended to ACP due to lack of novelty and truly new findings, however, does not state where findings of our work is previously published. The specific conclusions of this work were not published earlier thus provide unique results. In response to why we did not just use a 3D model, the box model is constrained to observed meteorological parameters and chemical species such as O3, NOx, CO, and some VOCs, which we find to be more useful than a 3D model for this kind of analysis since it eliminates some uncertainties, or errors that a 3D model could have. Our box model simulation could reduce uncertainties in the ozone production and sensitivity calculations.

We have stated at the end of Section 2.2: "The box model analysis is necessary for ozone production and its sensitivity to NOx and VOCs because the box model was constrained to measured species (e.g., NO, NO2, CO, HCHO, etc.) and meteorological parameters (e.g., photolysis frequencies) that are essential to calculate ozone production rates. Even though there is good agreement in general between the box model and the 3D model, there are still some differences between the measurements and the output from the 3D model, e.g., NOx, CO, HCHO and photolysis frequencies."

2. With regard to figures, Figure 1 is not necessary (the ozone isopleth is "classic"), Figure 2 would be better as a map with points/labels as the extraneous stuff is distracting, and Figures 3 and 4 can be combined. In addition, some of the figures are intuitive based on previous work in Houston and other locations (5, 6, 8, and 9).

Response: We would like to keep Figure 1 in the paper. Since Figure 1 is ozone production and not ozone concentration as traditional EKMA O3 isopleth diagrams are, it could provide useful information for the reader about how ozone production changes

with regarding to NOx and VOC and NOx and VOC sensitive regimes of ozone production. As suggested, we have changed Figure 2 to a map with points and labels. Figures 3 and 4 are combined. Figures 5, 6, 8 and 9 are the results from the DISCOVER-AQ Houston campaign showing spatial and temporal variations of ozone production and its sensitivity to NOx and VOCs. To our knowledge, there has not been a single study that covers such a large spatial range on this topic, and the data from this campaign provide us the unique opportunity to do such an analysis.

3. My largest criticism of this work is that it is known from three previous field campaigns that ozone production rates and sensitivities in Houston are temporally and spatially dependent. It seems to be that the most new information appears on lines 203-205 (line 206 is intuitive) regarding O3 loss and the split between RO2 and HO2 reactions with NO (unless this information is published elsewhere and I am unaware) and on line 255+ where it is noted that OPE has decreased in Houston compared to previous campaigns (due to the decrease in NOx emissions). I do not believe that these warrant a manuscript by themselves.

Response: The reviewer was right that there have been some previous studies, including three previous studies in Houston in 2000, 2006, and 2009 and some others in other locations, on ozone production and its relationships to NOx and VOCs (e.g., Kleinman et al., 2002; Ryerson et al., 2003; Newman et al., 2009; Mao et al., 2010; Chen et al., 2010; Ren et al., 2013), but to our knowledge, none of them has done systematic analysis on ozone production and its sensitivity to NOx and VOCs and covers such large spatial (urban and suburban) and temporal ranges as the DISCOVER-AQ Houston campaign does in 2013. For example, the SHARP study in 2009 (Ren et al., 2013) and the Texas Air Quality Study Radical and Aerosol Measurement Project (TRAMP) in 2006 (Mao et al., 2010; Chen et al., 2010) did cover ozone production and its sensitivity to NOx and VOCs, but they were focus on the data collected at a single location at Moody Tower at the University of Houston. Kleinman et al. (2002) and Ryerson et al., (2003) from TexAQS I in 2000 and Newman et al. (2009) from TexAQS II in

2006 discussed ozone production efficiencies (OPE), but they did not talk about the dependence of OPE on NOx and did not cover the sensitivity of ozone production to NOx and VOCs. The rich data set collected during the DISCOVER-AQ Houston campaign provides us a unique opportunity to perform this systematic analysis and we believe it is worth to inform the atmospheric chemistry community about the latest findings from this study to reflect the changes in chemical conditions (e.g., emissions) in Houston since previous studies.

4. The authors do not put Houston in the context of other locations. For example, they state on line 68 that "there are a limited number of observation-based studies on ozone production and its sensitivity to NOx and VOCs." There have been such studies made in Houston (SHARP, TEXAQS I and II) as well as in other locations across the US (Nashville, New England) and Europe. It would be appropriate to make such comparisons.

Response: We have cited results from other studies in other locations (e.g., Zaveri et al., 2003; Griffin et al., 2004; Thielmann et al., 2002) in Introduction and compared the results from this study to those from other locations. Our study is unique in that it examines the spatial and temporal variations in ozone production and its sensitivity. Other studies are mostly ground-based (i.e., single location like SHARP) or with limited spatial/temporal coverage. We found a higher OPE in this study than what was found in previous studies in Houston, which is probably due to continuous emission control as NOx levels were continuously pushed to ∼1ppbv and thus we got a higher OPE.

We have revised this sentence as: "There are some previous observation-based studies on ozone production and its relationships with NOx and VOCs (e.g., Thielmann et al., 2002; Zaveri et al., 2003; Ryerson et al., 2003; Griffin et al., 2003; Kleinman et al., 2005a; Neuman et al., 2009; Mao et al., 2010; Ren et al., 2013)." In Section 3.1, we have added one sentence: "Similar instantaneous ozone production rates have been observed in two previous studies in Houston in 2000 and 2006 [Kleinman et al., 2002a; Mao et al., 2010]." In Section 3.2, we revised a sentence to: "This OPE value is about

a factor of 1.5 to 2 higher than the OPEs obtained in the DISCOVER-AQ 2011 study in Maryland ranging from 4 to 5.5 (Ren, X., unpublished data), due to higher photochemical reactivity in Houston (Figure S4), but similar to 7.7-9.7 obtained from a ground site during the New England Air Quality Study (NEAQS) 2002 (Griffin et al., 2004)."

5. What is the basis for assuming a two-day lifetime for all calculated species to avoid build up?

Response: We do not provide a citation because we chose this value somewhat arbitrarily. By decreasing or increasing two days to one or ten days, it would not have much affect on the simulation results. This is because the box model already constrained all measured long-lived measured species. The additional lifetime of two days for the calculated species is to account for losses due to dry and wet deposition, vertical and horizontal diffusion, and to prevent accumulation of long-lived species in the box model. Most calculated species like OH, HO2 and RO2 are reactive intermediates and have lifetimes on the order of seconds to minutes, much shorter than 2 days. By adding this additional two-day lifetime would not affect the model results at all. There are a few long-lived species (like organic acid and alcohols) calculated in the model that could potentially accumulate to levels much higher than the levels in the ambient air.

We have revised this sentence: "An additional lifetime of two days was assumed for some calculated long lived species such as organic acids and alcohols to avoid unexpected accumulation of these species in the model."

Additional References

Griffin, R. J., C. A. Johnson, R. W. Talbot, H. Mao, R. S. Russo, Y. Zhou, and B. C. Sive (2004), Quantification of ozoneformation metrics at Thompson Farm during the New England Air Quality Study (NEAQS) 2002, J. Geophys. Res., 109, D24302,doi:10.1029/2004JD005344.

Ryerson, T. B., et al., Effect of petrochemical industrial emissions of reactive alkenes and NOx on tropospheric ozoneformation in Houston, Texas, J. Geophys. Res., 108(D8), 4249, doi:10.1029/2002JD003070, 2003.

Thielmann, A., A. S. H. Pre′voËEt, and J. Staehelin, Sensitivity of ozone production derived from field measurements in theItalian Po basin, J. Geophys. Res., 107(D22), 8194, doi:10.1029/2000JD000119, 2002.

Zaveri, R. A., C. M. Berkowitz, L. I. Kleinman, S. R. Springston, P. V. Doskey, W. A. Lonneman, and C. W. Spicer, Ozone production efficiency and NOx depletion in an urban plume: Interpretation of field observations and implications for evaluating O3-NOx-VOC sensitivity, J. Geophys. Res., 108(D14), 4436, doi:10.1029/2002JD003144, 2003.

Please also note the supplement to this comment:
http://www.atmos-chem-phys-discuss.net/acp-2016-215/acp-2016-215-AC2-supplement.pdf

[Figure]

[Figure]

**Fig. 1.** Figure 1 caption (2 in paper). DISCOVER-AQ ground and spiral sites (yellow dots) during the September 2013 Houston campaign.

**Supplement:**

Supporting Information for:

**Ozone Production and Its Sensitivity to NO$_x$ and VOCs: Results from the DISCOVER-AQ Field Experiment, Houston 2013**

Gina M. Mazzuca[1], Xinrong Ren[1,2,*], Christopher P. Loughner[2,3] Mark Estes[5], James H. Crawford[6], Kenneth E. Pickering[1,4], Andrew J. Weinheimer[7], and Russell R. Dickerson[1]

[1]Department of Atmospheric and Oceanic Science, University of Maryland, College Park, MD 20742, USA

[2]Air Resources Laboratory, National Oceanic and Atmospheric Administration, College Park, MD 20740, USA

[3]Earth System Science Interdisciplinary Center, University of Maryland, College Park, MD 20740, USA

[4]NASA Goddard Space Flight Center, Greenbelt, MD 20771, USA

[5]Texas Commission on Environmental Quality, Austin, TX 78711, USA

[6]NASA Langley Research Center, Hampton, VA 23681, USA

[7]National Center for Atmospheric Research, Boulder, CO 80307, USA

*Correspondence to: X. Ren (ren@umd.edu)

SI 1.

Both NOx and VOC levels in Houston have been continuously decreasing in the past 15-20 years as shown in Figure 1, the time series of NOx, ethane, and propene at two monitoring sites near the Houston Ship Channel.

[Figure]

[Figure]

**Figure S1.** Time series of NO, NOx, ethane and propene concentrations at the Deer Park and Clinton sites from 1998 to 2014. The Deer Park site is located in southeast of the Ship Channel. The Clinton site is located on the northwestern end of the Ship Channel. Each data point represents an average of hourly samples collected between July 1 and November 30 for each year. Missing data points indicate that too few valid samples (< 70%) were collected during that year. NO and NOx* data collected hourly using chemiluminescence sampler with molybdenum catalyst to convert NOx* (not true NOx because Mo catalyst converts other N species besides NO2 to NO) to NO. VOC data collected over a 40 minute period each hour using automated gas chromatography with cryogenic pre-concentration.

The NOx levels and OH reactivity in Houston during DAQ2013 and in Maryland during DAQ2011 are quite different, as shown in Figure 2. Houston has much higher NOx levels throughout the day. For OH reactivity, it is greater in Houston than in Maryland in the morning, but is comparable in both location in the afternoon. Note as shown in Figure 4, due to different emission sources, in Houston anthropogenic VOCs are the main contributor to the OH reactivity from VOCs, while in Maryland, biogenic VOCs (mainly isoprene) dominates the OH reactivity from VOCs. Different NOx levels and different VOC sources in Houston and Maryland are responsible for the different OPE values in the two areas.

[Figure]

**Figure S2.** Diurnal variations of NOx (left) and OH reactivity (Right) in Houston (linked blue circles) during DAQ2013 and in Maryland (linked red triangles) during DAQ2011.

SI 2.

CMAQ simulated a high bias in surface and aloft ozone (Table 1). CMAQ also simulated a low bias in CO, $CH_2O$, isoprene, $NO_2$, and NO aloft and a high bias in NOy aloft (Table 2). Recent work has shown that oceanic emissions of iodine and bromine result in ozone destruction (Carpenter et al., 2013). The high ozone bias in our results is expected due to the lack of oceanic iodine and bromine emissions and the associated chemistry. Biases in surface ozone are larger near the coastline (i.e., Galveston) than sites inland (i.e., Conroe).

**Table S1.** Mean bias (MB), normalized mean bias (NMB), normalized mean error (NME), root mean square error (RMSE), and Gross Error (GE) of surface ozone for the 2[nd] iterative 1 km WRF simulations covering all of September 2013.

|  | **Surface Ozone (ppbv)** |
| --- | --- |
| MB | 9.5 |
| NMB (%) | 39 |
| NME (%) | 51 |
| RMSE | 15 |
| GE | 12 |

**Table S2.** Second iterative 1 km CMAQ simulated mean bias (MB), normalized mean bias (NMB), normalized mean error (NME), and root mean square error (RMSE) of $O_3$, CO, $CH_2O$, Isoprene (ISO), $NO_2$, NO, and NOy covering measurements made onboard the NASA P-3B aircraft on all flight days during the DISCOVER-AQ field campaign

| | | $O_3$ | CO | $CH_2O$ | ISO | NO2 | NO | NOy |
|---|---|---|---|---|---|---|---|---|
| Model | MB | 0.8 | -5.8 | -0.3 | -0.02 | -0.5 | -0.3 | 0.04 |
| | NMB | 1.4 | -4.8 | -16 | -7.7 | -39 | -66 | 1.3 |
| | NME | 15 | 17 | 37 | 70 | 70 | 84 | 61 |
| | RMSE | 12 | 35 | 1.4 | 0.7 | 3.1 | 2.2 | 4.7 |

SI 3.

An evaluation of the improved WRF and CMAQ model simulations for the entire month of September 2013 was conducted. Statistics used to evaluate WRF and CMAQ are described Table S3. CMAQ simulated a high bias in surface and aloft ozone (Table S1). CMAQ also simulated a low bias in CO, $CH_2O$, isoprene, $NO_2$, and NO aloft and a high bias in NOy aloft (Table S2). Recent work has shown that oceanic emissions of iodine and bromine result in ozone destruction. The high ozone bias in our results is expected due to the lack of oceanic iodine and bromine emissions and the associated chemistry. Biases in surface ozone are larger near the coastline (i.e., Galveston) than sites inland (i.e., Conroe) as shown in Figure S3.

**Table S3.** Definition of the statistics used in WRF and CMAQ model evaluations. In these equations M represents the model results, O represents the observations, and N is the number of data points.

| Mean Bias (MB) | $MB = \dfrac{1}{N}\sum_{i=1}^{N}(M_i - O_i)$ |
|---|---|
| Normalized Mean Bias (NMB) | $NMB = \dfrac{\sum_{i=1}^{N}(M_i - O_i)}{\sum_{i=1}^{N} O_i} \times 100\%$ |
| Normalized Mean Error (NME) | $NME = \dfrac{\sum_{i=1}^{N}|M_i - O_i|}{\sum_{i=1}^{N} O_i} \times 100\%$ |
| Root Mean-Square Error (RMSE) | $RMSE = \sqrt{\dfrac{1}{N}\sum_{i=1}^{N}(M_i - O_i)^2}$ |
| Gross Error (G) | $GE = \dfrac{1}{N}\sum_{i=1}^{N}|M_i - O_i|$ |

**Table S4.** Mean bias (MB), normalized mean bias (NMB), normalized mean error (NME), root mean square error (RMSE), and Gross Error (GE) of 2 m temperature, 10 m wind speed, and 10 m wind direction for the 2[nd] iterative 1 km WRF simulations covering all of September 2013.

| | 2 m Temperature (K) | | 10 m Wind Speed (m/s) | | 10 m Wind Direction (deg) | |
|---|---|---|---|---|---|---|
| | | Model | | Model | | Model |
| MB | | 0.2 | | -0.8 | | 32 |
| NMB (%) | | 0.1 | | -17 | | 26 |
| NME (%) | | 0.4 | | 36 | | 26 |
| RMSE | | 1.6 | | 2.3 | | 51 |
| GE | | 1.2 | | 1.7 | | 32 |

[Figure]

**Figure S3.** Observed (*) and CMAQ simulated (solid lines) maximum 8 hour average ozone at La Porte Sylvan Beach (red), Conroe (purple), Galveston (blue), and West Houston (green) during September 2013.

SI 4.

The median OH reactivity due to non-methane hydrocarbons (NMHCs) was 3.3 s$^{-1}$ observed during DISCOVER-AQ 2013 in Houston and 1.2 s$^{-1}$ observed during DISCOVER-AQ 2011 in Maryland. As shown in Figure 2, alkanes and alkenes were dominant contributors to the OH reactivity due to NMHCs in Houston in 2013, while isoprene and alkanes were dominant contributors to the OH reactivity due to NMHCs in Maryland in 2011. The differences in overall OH reactivity and its distributions in the two locations are responsible to the different OPEs in the two different environments.

[Figure]

**Figure S4.** Distributions of OH reactivity due to non-methane hydrocarbons in DISCOVER-AQ 2011 in Maryland (left) and 2013 in Houston (right).

---

## Author Response (AR1)

Response to Anonymous Referee #1:

We thank the reviewer for providing insightful comments and helpful suggestions that have substantially improved the manuscript.  Below we have included the review comments followed by our responses in italic.  In the revision of this manuscript, we have highlighted those changes accordingly in blue font.

1) Review of "Ozone production and its sensitivity to $NO_X$ and VOCs: results from the DISCOVER-AQ field experiment, Houston 2013" The authors state several times that these results have important emissions control policy implications but it is not clear what type of program implementation would be needed based on the diurnal ozone production efficiencies presented here.

*Response: We are not suggesting a specific implementation program (which is beyond the scope of this work), however, are suggesting that it may be more beneficial at certain locations, during certain times of day, to regulate VOCs based on the diurnal ozone production efficiencies we report. We are providing a scientific basis through which policy makers could develop an emission reduction strategy.*

2) Given that this paper is focused on $NO_X$ and VOC contribution to $O_3$ production the authors should provide $NO_X$ and VOC measurements from this study and also compare those with previous Houston field studies to provide more context about how these pollutants are decreasing and for VOC how total VOC and VOC reactivity is decreasing to support conclusions about ozone production efficiency. Also, a comparison with another area like Baltimore would be useful.

*Response: Both NOx and VOC levels in Houston have been continuously decreasing in the past 15-20 years as shown in Figure 1(S1 in paper), the time series of NO, NOx, ethene, and propene at two monitoring sites near the Houston Ship Channel.*

[Figure]

[Figure]

**Figure 1.** Time series of NO, NOx, ethene and propene concentrations at the Deer Park and Clinton sites from 1998 to 2014. The Deer Park site is located southeast of the Ship Channel. The Clinton site is located on the northwestern end of the Ship Channel. Each data point represents an average of hourly samples collected between July 1 and November 30 for each year. Missing data points indicate that too few valid samples (< 70%) were collected during that year. NO and NOx* data collected hourly using chemiluminescence sampler with molybdenum catalyst to convert NOx* (not true NOx because Mo catalyst converts other N species besides $NO_2$ to NO) to NO. VOC data collected over a 40-minute period each hour using automated gas chromatography with cryogenic pre-concentration.

*The NOx levels and OH reactivity in Houston during DAQ2013 and in Maryland during DAQ2011 are quite different, as shown in Figure 2. Houston has much higher NOx levels throughout the day. For OH reactivity, it is greater in Houston than in Maryland in the morning, but is comparable in both locations in the afternoon. Note as shown in Figure 4, due to different emission sources, in Houston anthropogenic VOCs are the main contributor to the OH reactivity from VOCs, while in Maryland, biogenic VOCs (mainly isoprene) dominates the OH reactivity from VOCs. Different NOx levels and different VOC sources in Houston and Maryland are responsible for the different OPE values in the two areas.*

[Figure]

**Figure 2.** Diurnal variations of NOx (left) and OH reactivity (right) in Houston (linked blue circles) during DAQ2013 and in Maryland (linked red triangles) during DAQ2011.

3) The authors provide CMAQ simulated ozone production efficiency but provide no information about the emission inventory used for the simulation and how well the model predicted $NO_X$, $NO_Z$, VOC, and O3 compared with the aircraft and surface measurements made during the field study. Is it ok that the model predicts a similar OPE to the box model but not capture the magnitudes of the precursors or ozone correctly? The information presented about OPE is useful, but additional work is needed for this to provide a more comprehensive understanding of ozone production in Houston with respect to the models used by regulators for decision support and context from the many previous Houston field studies.

*Response: The WRF and CMAQ model options are described in Table 1. In Section 2.3, we also added the following a few sentences to describe the emissions we used in the CMAQ simulations: "The 2012 baseline anthropogenic emissions from the Texas Commission on Environmental Quality (TCEQ) were used as input to CMAQ. These emissions contain the most-up-to-date Texas anthropogenic emissions inventory and a compilation of emissions estimates from Regional Planning Offices throughout the US. Biogenic emissions were calculated online within CMAQ with Biogenic Emission Inventory System (BEIS). Lightning emissions were also calculated online within CMAQ."*

*CMAQ simulated a high bias in surface and aloft ozone (Tables 1). CMAQ also simulated a low bias in CO, $CH_2O$, isoprene, $NO_2$, and NO aloft and a high bias in NOy aloft (Table 2). Recent work has shown that oceanic emissions of iodine and bromine result in ozone destruction (Carpenter et al., 2013). The high ozone bias in our results is expected due to the lack of oceanic iodine and bromine emissions and the associated chemistry. Biases in surface ozone are larger near the coastline (i.e., Galveston) than sites inland (i.e., Conroe).*

**Table 1.** Mean bias (MB), normalized mean bias (NMB), normalized mean error (NME), root mean square error (RMSE), and Gross Error (GE) of surface ozone for the 2nd iterative 1 km WRF simulations covering all of September 2013.

|  | Surface Ozone (ppbv) |
|---|---|
| MB | 9.5 |
| NMB (%) | 39 |
| NME (%) | 51 |
| RMSE | 15 |
| GE | 12 |

**Table 2.** Second iterative 1 km CMAQ simulated mean bias (MB), normalized mean bias (NMB), normalized mean error (NME), and root mean square error (RMSE) of $O_3$, CO, $CH_2O$, Isoprene (ISO), $NO_2$, NO, and NOy covering measurements made onboard the NASA P-3B aircraft on all flight days during the DISCOVER-AQ field campaign.

|  |  | $O_3$ | CO | $CH_2O$ | ISO | NO2 | NO | NOy |
|---|---|---|---|---|---|---|---|---|
| **Model** | MB | 0.8 | -5.8 | -0.3 | -0.02 | -0.5 | -0.3 | 0.04 |
| | NMB | 1.4 | -4.8 | -16 | -7.7 | -39 | -66 | 1.3 |
| | NME | 15 | 17 | 37 | 70 | 70 | 84 | 61 |
| | RMSE | 12 | 35 | 1.4 | 0.7 | 3.1 | 2.2 | 4.7 |

4) The last half of the introduction section reads like a white paper on the Houston DISCOVER-AQ field study. Since this paper does not present any information relevant to the mission of that field study which was to validate satellite measurements the discussion of the DISCOVER-AQ campaign could be de-emphasized in favor of more time spent on the multitude of historical field studies in the Houston area. Also, the authors never clearly state in the introduction what they are presenting and why that information is novel.

*Response: We have removed lines 89-96 and combine lines 97 – 100 and took out lines 102-106. We edited lines 81-84 to read: "In the work presented here, we provide investigations of spatial and temporal variations of ozone production and its sensitivity to NOx and VOCs to provide a scientific basis to develop a non-uniform emission reduction strategy for $O_3$ pollution control in urban areas such as Houston."*

5) The authors do not need to explain why CB05 is used rather than CBIV, but an explanation about why CB05 was used rather than the newer version CB6 is necessary. At several points in the manuscript the authors note than organic nitrate fate can confound OPE interpretation so the choice of an older Carbon Bond mechanism that has a less realistic treatment of organic nitrates is needed. Also, it is not clear why all species have the same two-day deposition lifetime. Species like O3 and HNO3 deposit out of the atmosphere and very different rates.

*Response: CB05 is the most up to date Carbon Bond mechanism in CMAQ (i.e., CB6 has not been implemented into CMAQ at the time the analysis was performed). The box model was constrained for all long-lived measured species like ozone and HNO$_3$ and we do not assume a two-day deposition lifetime. An additional two-day lifetime due to deposition and heterogeneous losses is assumed for calculated species in the box model. Most calculated species like OH, HO$_2$ and RO$_2$ are reactive intermediates and have lifetimes on the order of seconds to minutes, much shorter than 2 days. Adding this additional two-day lifetime would not affect the model results at all. There are a few long-lived species (like organic acid and alcohols) calculated in the model that could potentially accumulate to levels much higher than the levels in the ambient air. We have revised this sentence: "An additional lifetime of two days was assumed for some calculated long lived species such as organic acids and alcohols to avoid unexpected accumulation of these species in the model."*

6) Please provide information about the emission inventory and modeling used as input to the CMAQ simulation and the source of the initial and boundary conditions.

*Response: The WRF and CMAQ model options have been described in Table 1. In Section 2.3, we also added the following a few sentences for the emissions we used in the CMAQ simulations: "The 2012 baseline anthropogenic emissions from the Texas Commission on Environmental Quality (TCEQ) were used as input to CMAQ. These emissions contain the most-up-to-date Texas anthropogenic emissions inventory and a compilation of emissions estimates from Regional Planning Offices throughout the US Biogenic emissions was calculated online within CMAQ with Biogenic Emission Inventory System (BEIS). Lightning emissions were also calculated online within CMAQ." It is also listed in Table 1 of this manuscript.*

7) In the results section, please provide some comparison of CMAQ estimated VOC, speciated VOC, NO, NO$_2$, HNO3, PANs, HNO3, and O3 with measurements.

*Response: An evaluation of the improved WRF and CMAQ model simulations for the entire month of September 2013 was conducted. Statistics used to evaluate WRF and CMAQ are described Tables 3. CMAQ simulated a high bias in surface and aloft ozone (Table 1). CMAQ also simulated a low bias in CO, CH$_2$O, isoprene, NO$_2$, and NO aloft and a high bias in NOy aloft (Table 2). Recent work has shown that oceanic emissions of iodine and bromine result in ozone destruction. The high ozone bias in our results is expected due to the lack of oceanic iodine and bromine emissions and the associated chemistry. Biases in surface ozone are larger near the coastline (i.e., Galveston) than sites inland (i.e., Conroe) as shown in Figure 3.*

**Table 3.** Definition of the statistics used in WRF and CMAQ model evaluations. In these equations M represents the model results, O represents the observations, and N is the number of data points.

| Mean Bias (MB) | $$MB = \frac{1}{N}\sum_{i=1}^{N}(M_i - O_i)$$ |
|---|---|
| Normalized Mean Bias (NMB) | $$NMB = \frac{\sum_{i=1}^{N}(M_i - O_i)}{\sum_{i=1}^{N}O_i} \times 100\%$$ |
| Normalized Mean Error (NME) | $$NME = \frac{\sum_{i=1}^{N}|M_i - O_i|}{\sum_{i=1}^{N}O_i} \times 100\%$$ |
| Root Mean-Square Error (RMSE) | $$RMSE = \sqrt{\frac{1}{N}\sum_{i=1}^{N}(M_i - O_i)^2}$$ |
| Gross Error (G) | $$GE = \frac{1}{N}\sum_{i=1}^{N}|M_i - O_i|$$ |

**Table 4.** Mean bias (MB), normalized mean bias (NMB), normalized mean error (NME), root mean square error (RMSE), and Gross Error (GE) of 2 m temperature, 10 m wind speed, and 10 m wind direction for the 2nd iterative 1 km WRF simulations covering all of September 2013.

| | 2 m Temperature (K) | | 10 m Wind Speed (m/s) | | 10 m Wind Direction (deg) | |
|---|---|---|---|---|---|---|
| | | Model | | Model | | Model |
| MB | | 0.2 | | -0.8 | | 32 |
| NMB (%) | | 0.1 | | -17 | | 26 |
| NME (%) | | 0.4 | | 36 | | 26 |
| RMSE | | 1.6 | | 2.3 | | 51 |
| GE | | 1.2 | | 1.7 | | 32 |

[Figure]

**Figure 3.** Observed (\*) and CMAQ simulated (solid lines) maximum 8 hour average ozone at La Porte Sylvan Beach (red), Conroe (purple), Galveston (blue), and West Houston (green) during September 2013.

8) The authors suggest one difference in OPE between Houston and Baltimore is due to reactivity. Please provide speciated VOC concentrations from each field study by reactivity so this relationship is clearer.

*Response: The median OH reactivity due to non-methane hydrocarbons (NMHCs) was 3.3 s⁻¹ observed during DISCOVER-AQ 2013 in Houston and 1.2 s⁻¹ observed during DISCOVER-AQ 2011 in Maryland. As shown in Figure 4, alkanes and alkenes were dominant contributors to the OH reactivity due to NMHCs in Houston in 2013, while isoprene and alkanes were dominant contributors to the OH reactivity due to NMHCs in Maryland in 2011. The differences in overall OH reactivity and its distributions in the two locations are responsible to the different OPEs in the two different environments. We have included this in the Supporting Information.*

[Figure]

**Figure 4.** Distributions of OH reactivity due to non-methane hydrocarbons in DISCOVER-AQ 2011 in Maryland (left) and 2013 in Houston (right).

9) The authors make a lot of strong conclusions about trends in OPE when NOX is greater or less than 1 ppb as shown in Figure 14. The points in Figure 14 do not show a distinct relationship above or below any level of the NOX concentrations. Perhaps box plots binned by NOX concentration would be a better way to show this type of relationship (if it really exists).

*Response: We have updated Figure 13 by adding median OPE values binned by NOx concentration on top of the individual data points and the trend seems more distinct.*

[Figure]

**Figure 13.** Ozone production efficiency (OPE) versus NOx in the box model (blue circles) and the CMAQ model pink dots) results. The linked blue circles show the median OPE values binned by NOx concentration in the box model, while the linked red triangles show the median OPE values binned by NOx concentration in the CMAQ model, OPE is calculated according to its definition as the net ozone formation rate divided by of the formation rate of NOz.

Response to Anonymous Referee #2:

We thank the reviewer for providing insightful comments and helpful suggestions that have substantially improved the manuscript. Below we have included the review comments followed by our responses in italic. In the revision of this manuscript, we have highlighted those changes accordingly in blue font.

1. The analyses performed and the approach used are tried and true so technically, there are no major faults with the work (though I question the use of a box model in Houston when the meteorology is so complex - why not just use the 3D model as it can provide answers to some of the questions asked and the ambient data can be used for model evaluation). However, due to a lack of novelty and a lack of truly new findings that warrant an entire manuscript, I am unable to recommend this manuscript for publication in ACP.

Response: *The reviewer's comment prompted us to re-examine the literature, where we found a few more relevant papers (i.e. Thielmann et al., 2002; Zaveri et al., 2003; Ryerson et al., 2003; Griffin et al., 2003, Kommalapati et al., 2016), but none that thoroughly addressed the issues that we cover in this paper.*

*In response to why we did not just use a 3D model, the box model is constrained to observed meteorological parameters and chemical species such as $O_3$, $NO_x$, CO, and some VOCs, which we find to be more useful than a 3D model for this kind of analysis since it eliminates some uncertainties, or errors that a 3D model could have. A 3D CTM may have major problems with the emissions inventories as described by Yu et al. (2012) and Travis et al. (in review in ACPD, 2016), who show that modeled NOy was twice as high than observed. Our box model simulation could reduce uncertainties in the ozone production and sensitivity calculations.*

*We have stated at the end of Section 2.2: "The box model analysis is necessary for ozone production and its sensitivity to $NO_x$ and VOCs because the box model was constrained to measured species (e.g., NO, $NO_2$, CO, HCHO, etc.) and meteorological parameters (e.g., photolysis frequencies) that are essential to calculate ozone production rates. Even though there is good agreement in general between the box model and the 3D model, there are still some differences between the measurements and the output from the 3D model, e.g., NOx, CO, HCHO and photolysis frequencies."*

2. With regard to figures, Figure 1 is not necessary (the ozone isopleth is "classic"), Figure 2 would be better as a map with points/labels as the extraneous stuff is distracting, and Figures 3 and 4 can be combined. In addition, some of the figures are intuitive based on previous work in Houston and other locations (5, 6, 8, and 9).

*Response: We would like to keep Figure 1 in the paper. Since Figure 1 is ozone production and not ozone concentration as traditional EKMA $O_3$ isopleth diagrams are, it could provide useful information for the reader about how ozone production changes with regarding to NOx and VOC and NOx and VOC sensitive regimes of ozone production. As suggested, we have changed*

*Figure 2 to a map with points and labels. Figures 3 and 4 are combined. Figures 5, 6, 8 and 9 are the results from the DISCOVER-AQ Houston campaign showing spatial and temporal variations of ozone production and its sensitivity to NOx and VOCs. To our knowledge, there has not been a single study that covers such a large spatial range on this topic, and the data from this campaign provide us the unique opportunity to do such an analysis.*

[Figure]

**Figure 2.** DISCOVER-AQ ground and spiral sites (yellow dots) during the September 2013 Houston campaign.

3. My largest criticism of this work is that it is known from three previous field campaigns that ozone production rates and sensitivities in Houston are temporally and spatially dependent. It seems to be that the most new information appears on lines 203-205 (line 206 is intuitive) regarding O3 loss and the split between RO2 and HO2 reactions with NO (unless this information is published elsewhere and I am unaware) and on line 255+ where it is noted that OPE has decreased in Houston compared to previous campaigns (due to the decrease in NOx emissions). I do not believe that these warrant a manuscript by themselves.

*Response: The reviewer was right that there have been some previous studies, including three previous studies in Houston in 2000, 2006, and 2009 and some others in other locations, on ozone production and its relationships to NOx and VOCs (e.g., Kleinman et al., 2002; Ryerson et al., 2003; Newman et al., 2009; Mao et al., 2010; Chen et al., 2010; Ren et al., 2013), but to our*

*knowledge, none of them has done systematic analysis on ozone production and its sensitivity to NOx and VOCs and covers such large spatial (urban and suburban) and temporal ranges as the DISCOVER-AQ Houston campaign does in 2013. For example, the SHARP study in 2009 (Ren et al., 2013) and the Texas Air Quality Study Radical and Aerosol Measurement Project (TRAMP) in 2006 (Mao et al., 2010; Chen et al., 2010) did cover ozone production and its sensitivity to NOx and VOCs, but they were focus on the data collected at a single location at Moody Tower at the University of Houston. Kleinman et al. (2002) and Ryerson et al., (2003) from TexAQS I in 2000 and Newman et al. (2009) from TexAQS II in 2006 discussed ozone production efficiencies (OPE), but they did not talk about the dependence of OPE on NOx and did not cover the sensitivity of ozone production to NOx and VOCs. The rich data set collected during the DISCOVER-AQ Houston campaign provides us a unique opportunity to perform this systematic analysis and we believe it is worth to inform the atmospheric chemistry community about the latest findings from this study to reflect the changes in chemical conditions (e.g., emissions) in Houston since previous studies.*

4. The authors do not put Houston in the context of other locations. For example, they state on line 68 that "there are a limited number of observation-based studies on ozone production and its sensitivity to NOx and VOCs." There have been such studies made in Houston (SHARP, TEXAQS I and II) as well as in other locations across the US (Nashville, New England) and Europe. It would be appropriate to make such comparisons.

*Response: We have cited results from other studies in other locations (e.g., Zaveri et al., 2003; Griffin et al., 2004; Thielmann et al., 2002) in the introduction and compared the results from this study to those from other locations. Our study is unique in that it examines the spatial and temporal variations in ozone production and its sensitivity. Other studies are mostly ground-based (i.e., single location like SHARP) or with limited spatial/temporal coverage. We found a higher OPE in this study than what was found in previous studies in Houston, which is probably due to continuous emission control as NOx levels were continuously pushed to ~1ppbv and thus we got a higher OPE.*

*We have revised this sentence as: "There are some observation-based studies on ozone production and its relationships with $NO_x$ and VOCs [e.g., Thielmann et al., 2002; Zaveri et al., 2003; Ryerson et al., 2003; Griffin et al., 2003; Kleinman et al., 2005a; Neuman et al., 2009; Mao et al., 2010; Ren et al., 2013]"*

*In Section 3.1, we have added one sentence: "Similar instantaneous ozone production rates have been observed in two previous studies in Houston in 2000 and 2006 [Kleinman et al., 2002a; Mao et al., 2010]."*

*In Section 3.2, we revised a sentence to: "Houston area OPE values range from about a factor of 1.3 to 2 higher than the OPEs calculated from the DISCOVER-AQ 2011 study in Maryland, likely due to higher photochemical reactivity in Houston (Figure S4). The 2011 Maryland OPEs*

*ranged from 3.4 to 6.1 when all measured data below 1 km are used (Ren, X., unpublished data). An OPE of ~8 was calculated [He et al., 2013] for the 2011 Maryland DISCOVER-AQ campaign for measured data below the 850 hPa level during vertical spirals with a strong linear correlation ($r^2 > 0.8$) between $O_x$ and $NO_z$. Additionally, OPEs of 7.7-9.7 were obtained from a ground site during the New England Air Quality Study (NEAQS) 2002 (Griffin et al., 2004)."*

5. What is the basis for assuming a two-day lifetime for all calculated species to avoid build up?

*Response: We do not provide a citation because we chose this value somewhat arbitrarily. By decreasing or increasing two days to one or ten days, it would not have much affect on the simulation results. This is because the box model already constrained all measured long-lived measured species. The additional lifetime of two days for the calculated species is to account for losses due to dry and wet deposition, vertical and horizontal diffusion, and to prevent accumulation of long-lived species in the box model. Most calculated species like OH, $HO_2$ and $RO_2$ are reactive intermediates and have lifetimes on the order of seconds to minutes, much shorter than 2 days. By adding this additional two-day lifetime would not affect the model results at all. There are a few long-lived species (like organic acid and alcohols) calculated in the model that could potentially accumulate to levels much higher than the levels in the ambient air.*

*We have revised this sentence: "An additional lifetime of two days was assumed for some calculated long lived species such as organic acids and alcohols to avoid unexpected accumulation of these species in the model."*

**Additional References**

*Carpenter, L. J., S. M. MacDonald, M. D. Shaw, R. Kumar, R. W. Saunders, R. Parthipan, J. Wilson, and J. M. C. Plane (2013), Atmospheric iodine levels influenced by sea surface emissions of inorganic iodine, Nat Geosci, 6(2), 108-111.*

*Griffin, R. J., C. A. Johnson, R. W. Talbot, H. Mao, R. S. Russo, Y. Zhou, and B. C. Sive (2004), Quantification of ozone formation metrics at Thompson Farm during the New England Air Quality Study (NEAQS) 2002, J. Geophys. Res., 109, D24302,doi:10.1029/2004JD005344.*

*Kommalapati, R. R., Z. Liang, and Z. Huque (2016), Photochemical model simulations of air quality for Houston-Galveston-Brazoria area and analysis of ozone-NO (x) -hydrocarbon sensitivity, International Journal of Environmental Science and Technology, 13(1), 209-220.)*

*Ryerson, T. B., et al., Effect of petrochemical industrial emissions of reactive alkenes and NOx on tropospheric ozone formation in Houston, Texas, J. Geophys. Res., 108(D8), 4249, doi:10.1029/2002JD003070, 2003.*

*Thielmann, A., A. S. H. Preˊvoˆt, and J. Staehelin, Sensitivity of ozone production derived from*

*field measurements in theItalian Po basin, J. Geophys. Res., 107(D22), 8194, doi:10.1029/2000JD000119, 2002.*

*Yu, S. C., et al. (2012), Comparative evaluation of the impact of WRF-NMM and WRF-ARW meteorology on CMAQ simulations for O3 and related species during the 2006 TexAQS/GoMACCS campaign, Atmospheric Pollution Research, 3(2), 149-162.*

*Zaveri, R. A., C. M. Berkowitz, L. I. Kleinman, S. R. Springston, P. V. Doskey, W. A. Lonneman, and C. W. Spicer, Ozone production efficiency and NOx depletion in an urban plume: Interpretation of field observations and implications for evaluating O3-NOx-VOC sensitivity, J. Geophys. Res., 108(D14), 4436, doi:10.1029/2002JD003144, 2003.*

**List of changes in revised, *marked up* manuscript below: (line numbers may be slightly off on unmarked up version)**

1) We revised the sentences on lines 68-70 to read:

"There are some observation-based studies on ozone production and its relationships with $NO_x$ and VOCs [e.g., Thielmann et al., 2002; Zaveri et al., 2003; Ryerson et al., 2003; Griffin et al., 2003; Kleinman et al., 2005a; Neuman et al., 2009; Mao et al., 2010; Ren et al., 2013]"

2) We have removed lines 91-98 and combine lines 99 – 102 and took out lines 104-108. We edited lines 83-86 to read:

"In the work presented here, we provide investigations of spatial and temporal variations of ozone production and its sensitivity to NOx and VOCs to provide a scientific basis to develop a non-uniform emission reduction strategy for $O_3$ pollution control in urban areas such as Houston."

3) We have revised the sentence on lines 144-146 to read:
"An additional lifetime of two days was assumed for some calculated long-lived species such as organic acids and alcohols to avoid unexpected accumulation of these species in the model."

4) We added the following on lines 153-159:

"The box model analysis is necessary for ozone production and its sensitivity to $NO_x$ and VOCs because the box model was constrained to measured species (e.g., NO, $NO_2$, CO, HCHO, etc.) and meteorological parameters (e.g., photolysis frequencies) that are essential to calculate ozone production rates. Even though there is good agreement in general between the box model and the 3D model, there are still some differences between the measurements and the output from the 3D model, e.g., NOx, CO, HCHO and photolysis frequencies."

5) We added the following on lines: 170-176

"The 2012 baseline anthropogenic emissions from the Texas Commission on Environmental Quality (TCEQ) were used as input to CMAQ. These emissions contain the most-up-to-date Texas anthropogenic emissions inventory and a compilation of emissions estimates from Regional Planning Offices throughout the US. Biogenic emissions were calculated online within CMAQ with Biogenic Emission Inventory System (BEIS). Lightning emissions were also calculated online within CMAQ."

6) On lines 189-190 in Section 3.1, we have added one sentence:

"Similar instantaneous ozone production rates have been observed in two previous studies in Houston in 2000 and 2006 [Kleinman et al., 2002a; Mao et al., 2010]."

7) On lines 257-264 in Section 3.2, we revised a sentence to read:

 "Houston area OPE values range from about a factor of 1.3 to 2 higher than the OPEs calculated from the DISCOVER-AQ 2011 study in Maryland, likely due to higher photochemical reactivity in Houston (Figure S4). The 2011 Maryland OPEs ranged from 3.4 to 6.1 when all measured data below 1 km are used (Ren, X., unpublished data). An OPE of ~8 was calculated [He et al., 2013] for the 2011 Maryland DISCOVER-AQ campaign for measured data below the 850 hPa level during vertical spirals with a strong linear correlation ($r^2 > 0.8$) between $O_x$ and $NO_z$. Additionally, OPEs of 7.7-9.7 were obtained from a ground site during the New England Air Quality Study (NEAQS) 2002 (Griffin et al., 2004)."

8) We updated figure 2 (line 466)

9) We combined originally separate figures into Figure 3 (line 471)

10) We updated figure 8 (line 494)

11) We have updated Figure 13 by adding median OPE values binned by NOx concentration on top of the individual data points and the trend seems more distinct (line 530)

[revised manuscript text omitted]